# On the assimilation of environmental tracer observations for model-based decision support

Matthew J. Knowling[1], Jeremy T. White[1], Catherine R. Moore[1], Pawel Rakowski[2], and Kevin Hayley[3]

[1]GNS Science, New Zealand
[2]Hawke's Bay Regional Council, New Zealand
[3]Groundwater Solutions Ltd, Australia

**Correspondence:** Matthew J. Knowling (m.knowling@gns.cri.nz)

**Abstract.** It has been advocated that history-matching numerical models to a diverse range of observation data types, particularly including environmental tracer concentrations and their interpretations/derivatives (e.g., mean age), constitutes an effective and appropriate means to improve model forecast reliability. This study presents two regional-scale modeling case studies that directly and rigorously assess the value of discrete tritium concentration observations and tritium-derived mean residence time (MRT) estimates in two decision-support contexts; "value" is measured herein as both the improvement (or otherwise) in the reliability of forecasts through uncertainty variance reduction and bias minimization as a result of assimilating tritium or tritium-derived MRT observations. The first case study (Heretaunga Plains, New Zealand) utilizes a suite of steady-state and transient flow models and an advection-only particle-tracking model to evaluate the worth of tritium-derived MRT estimates relative to hydraulic potential, spring discharge and river/aquifer exchange flux observations. The worth of MRT observations is quantified in terms of the change in the uncertainty surrounding ecologically-sensitive spring discharge forecasts via first-order second-moment (FOSM) analyses. The second case study (Hauraki Plains, New Zealand) employs paired simple/complex transient flow and transport models to evaluate the potential for assimilation-induced bias in simulated surface-water nitrate discharge to an ecologically-sensitive estuary system; formal data assimilation of tritium observations is undertaken using an iterative ensemble smoother. The results of these case studies indicate that, for the decision-relevant forecasts considered, tritium observations are of variable benefit and may induce damaging bias in forecasts; these biases are a result of an imperfect model's inability to properly and directly assimilate the rich information content of the tritium observations. The findings of this study challenge the advocacy of the increasing use of tracers, and diverse data types more generally, whenever environmental model data assimilation is undertaken with imperfect models. This study also highlights the need for improved imperfect-model data assimilation strategies. While these strategies will likely require increased model complexity (including advanced discretization, processes and parameterization) to allow for appropriate assimilation of rich and diverse data types that operate across a range of spatial and temporal scales commensurate with a forecast of management interest, it is critical that increased model complexity does not preclude the application of formal data assimilation and uncertainty quantification techniques due to model instability and excessive run times.

# 1 Introduction

Numerical models used to provide water resources management decision support are often subjected to data assimilation through history matching (or "calibration"). This is due to the large information deficit accompanying the development of these models, and the potential for the history matching process to lead to an increased reliability of simulated outputs of management interest (herein referred to as "forecasts") through variance reduction. Modeling for the purpose of decision-support is the context in which the remainder of the paper is framed.

It is widely advocated that the assimilation of multiple types of state observations (i.e., "diverse data") is of benefit in "constraining" models. In other words, as more data are used for history matching, and the more diverse those data are, the reliability of the forecasts increases. This is an intuitive stance arising from direct application of Bayes equation and from the recognized rich information content of diverse data types; this intuition is supported by many studies, (e.g., Sanford et al., 2004; Michael and Voss, 2009; Ginn et al., 2009; Li et al., 2009; Gusyev et al., 2013; Hansen et al., 2013). For example, Hunt et al. (2006) demonstrated the importance of unconventional observations including lake/aquifer exchange fluxes, depth of lake isotope plume and groundwater travel times in achieving "well-constrained parameter values" (e.g., acceptable posterior variance) through history matching a regional-scale groundwater model.

History-matching to environmental tracer observations, in particular, is widely a regarded mechanism to improve the reliability of forecasts. In a review of approaches for modeling environmental tracers in groundwater systems, Turnadge and Smerdon (2014) state that age data have been useful for constraining models; in particular, "simulation of environmental tracer transport that explicitly accounts for the accumulation and decay of tracer mass, has proven to be highly beneficial in constraining numerical models". Zell et al. (2018) showed the relative importance of water-level, stream discharge and environmental tracers (including tritium, CFCs, SF6) in the conditioning of groundwater travel time forecasts. They reported that, overall, tracer data were of considerable benefit in terms of forecast uncertainty reduction. In a recent review paper, Schilling et al. (2019) state that assimilation of concentration observations through surface water/groundwater flow model history matching "harbors huge potential", based on the findings of previous studies, while assimilation of tracer-derived residence time observations in these models also often help significantly (where an appropriate approach is adopted, (e.g., Sanford, 2011; Zuber et al., 2011).

However, the extent to which the assimilation of diverse data types (including environmental tracers) is of benefit has previously been investigated only from a somewhat theoretical standpoint, i.e., neglecting the effects of model error. Direct evaluation of the likelihood term of Bayes theorem is predicated on a "perfect" simulator to appropriately condition uncertain model parameters through data assimilation. In real-world modeling contexts, however, the presence of model error can invalidate even the most rigorous data assimilation techniques (e.g., Doherty and Welter, 2010; White et al., 2014; Oliver and Alfonzo, 2018). Therefore, when an imperfect simulator is used in a data assimilation framework, extreme care must be taken to assure that the model imperfections do not corrupt (through biased first moments, or under-estimated second moments) the forecast posterior distributions. A number of recent works have shown that the failure to appropriately frame the imperfect-model data assimilation problem can result in severely biased results (e.g., Doherty and Christensen, 2011; Knowling et al., 2019; White et al., 2020).

The largely unknown ability of an imperfect regional-scale model to simultaneously assimilate diverse data types that operate over different spatial and temporal scales—and how these imperfections may affect model-based decision support in some contexts—serves as motivation for the current study. To the best of the authors' knowledge, this is the first study to explore the benefit or otherwise of the assimilation of tracer data into imperfect models in terms of both forecast bias and variance.

A subtle, yet very important distinction should be made at this point. There is no doubting that diverse data types, and in particular environmental tracers, have contributed significantly to the understanding of catchment processes and properties (e.g., Kirchner et al., 2001; André et al., 2005; Stewart and Thomas, 2008; McDonnell et al., 2010; Morgenstern et al., 2010; Han et al., 2012; Leray et al., 2012; Siade et al., 2018). However, as discussed, this study focuses instead on the role of (imperfect) models in two selected decision-support contexts, and how the assimilation of environmental tracers in particular affects their utility in these contexts, i.e., by increasing (or otherwise) the reliability of forecasts.

Herein, we focus specifically on the ramifications of assimilating the information contained within tritium concentration observations and tritium-derived mean residence time (MRT) observations for model-based decision support concerning low flow and nutrient transport at the regional scale in hydrological environments where young groundwater components are decision relevant. Tritium is a popular tracer for the identification of relatively young age groundwaters (i.e., <70 years old), for the following reasons: ($i$) unlike CFCs, tritium is not affected by microbial degradation or contamination; and ($ii$) unlike SF6, it is not affected by potential subsurface sources (e.g., Morgenstern and Daughney, 2012; Cartwright and Morgenstern, 2012; Beyer et al., 2014).

The objective of this study is two-fold. First, we investigate the theoretical worth of tritium-derived MRT observations relative to other observation data types. This investigation is performed using a case study (Heretaunga Plains, New Zealand) that adopts first-order second-moment (FOSM) techniques; our analysis focuses on the relative worth of MRT observations in terms of changes in the uncertainty associated with spring discharge forecasts at various locations that are of management interest due to their ecological significance. This first case study employs advective-only particle-tracking modeling approach to simulate MRT.

Second, we explore the use of discrete tritium concentration observations in data assimilation in the context of a controlled model simplification experiment as a means to understand what, if any, ill-effects may be induced by using these information-rich data types in a simplified (i.e., imperfect) model. This exploration is performed using a second case study that employs a recently-presented paired simple/complex model analysis (White et al., 2020). The paired model analysis is used herein to allow for the identification of possible (and otherwise undetectable) bias and uncertainty under-estimation surrounding forecasts of nutrient load to an ecologically-sensitive estuary system. This second case study simulates (tritium and nitrate) tracer concentrations directly—using a full advective-dispersive modeling approach that also accounts for first-order reaction rates.

## 2 First case study

The first case study serves to investigate the ability of tritium-derived MRT observations to constrain ecologically-sensitive spring discharge forecasts (i.e., the "worth" of these observations) using a model of the groundwater system of the Heretaunga Plains (New Zealand) (Figure 1). The model was constructed primarily for the purposes of groundwater allocation management decision-support.

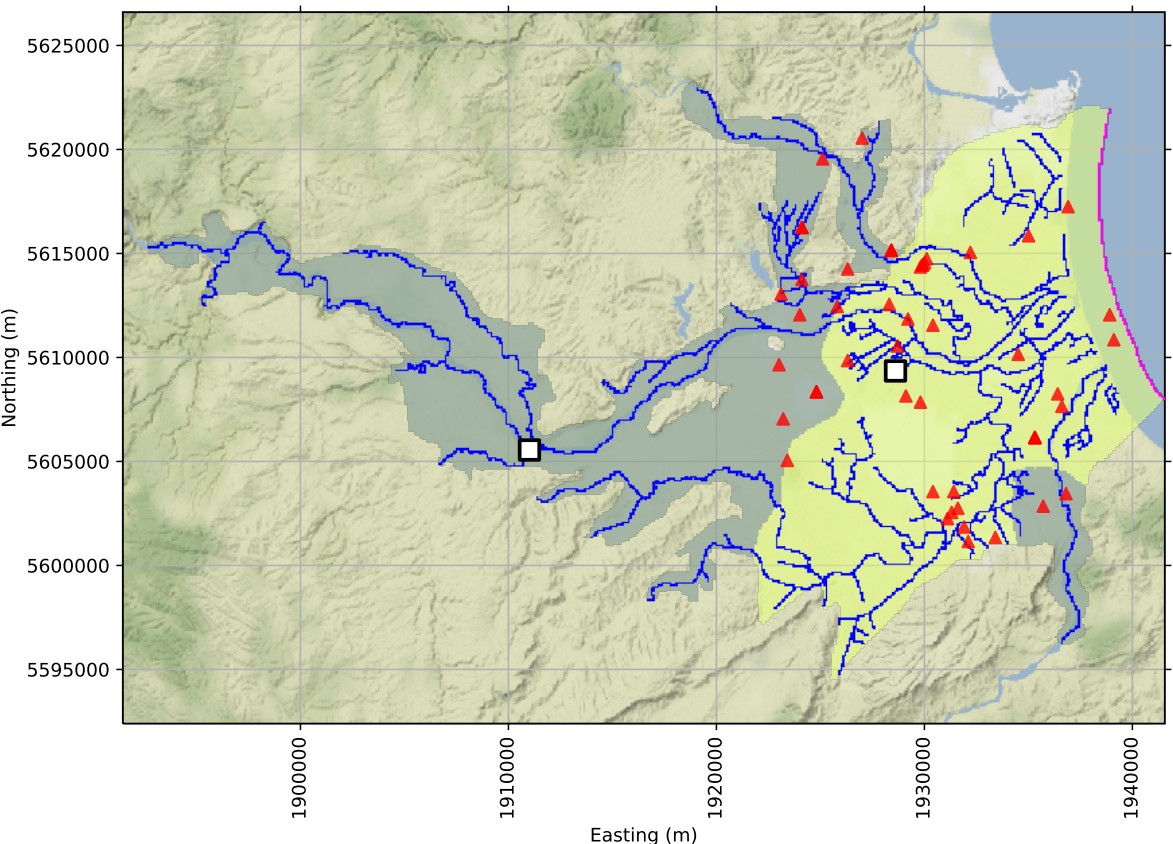

**Figure 1.** Heretaunga Plains model schematic, including river boundary conditions (blue lines), top-layer confinement status (unconfined areas shaded blue and confined areas shaded yellow), coastal general-head boundary (magenta line). The location of groundwater tritium-derived MRT observations are shown as red triangles. The location of forecasts—spring discharge rates during summer and winter—are shown as white markers.

### 2.1 The model

The model comprises 302 rows and 501 columns (uniform 100 m by 100 m horizontal grid discretization). Two layers are used for flow simulations, whereas six layers are used to generate more vertically detailed cell-by-cell flow budgets for par-

ticle tracking simulations. MODFLOW-2005 (Harbaugh, 2005) is used to simulate groundwater flow under steady-state and transient conditions. Separate simulations are conducted for data assimilation and forecasting purposes spanning different time periods (and temporal resolutions) of interest (e.g., separate transient flow simulations are conducted using annual stress periods for the period 1980—2015, and using monthly stress periods for the periods 1997—1999 and 2011—2015). MODPATH (Pollock, 2012) is used to simulate advection-only (i.e., neglecting diffusion, dispersion and retardation) reverse particle tracking, thereby providing a basis for assimilating tritium-derived MRT estimates (Figure 1). Specifically, the mean particle exit time corresponding to each observation location is compared with tritium-derived MRT estimates (e.g., Sanford, 2011; Gusyev et al., 2014).

Relevant aspects of the model are as follows:

- Land-surface recharge estimates, derived from a daily soil water balance modeling assessment (Rajanayaka and Fisk, 2018), are specified using the (specified flux) recharge package.

- The interaction between groundwater and surface water (including rivers, streams and springs) is simulated using the (head-dependent flux) river package. Time-varying river stage values are specified for the three main rivers in the region based on observed values. River-bed conductance values are varied seasonally to reflect in an approximate manner the non-linear relationship between field observations of spring discharge and groundwater levels.

- The coastal boundary condition is represented using the (head-dependent flux) general-head boundary package. The general-head stage is specified using a density-corrected mean sea-level (e.g., Morgan et al., 2012).

- Groundwater abstraction rates, based on observed and estimated data, are represented using the (specified flux) well package.

For a more detailed description of the Heretaunga Plains models, the reader is referred to Rakowski and Knowling (2018).

## 2.2   Forecasts

We focus on the following forecasts—due to their ecological significance and their potential to be impacted by groundwater abstraction:

- Spring discharge rate during summer at two locations (one in the central Heretaunga Plains, and one in the upper reaches of the catchment) (Figure 1)

- Spring discharge rate during winter at the central Heretaunga Plains location (Figure 1)

## 2.3   Observations for assimilation

Data assimilation is undertaken notionally via FOSM techniques using the following observations:

- 6,167 groundwater levels (comprising time-averaged water-levels, absolute and deviation-from-mean annual, monthly and daily water-levels, long-term differences in water-level, and vertical head differences);

- surface-water/groundwater fluxes (time-averaged and transient river gain and loss fluxes and spring discharge fluxes, obtained using a range of techniques including flow gauging, electrical conductivity and temperature surveys, water isotopic analyses, etc. (Wilding, 2017)); and

130
- 52 groundwater MRT estimates derived from tritium concentrations using lumped-parameter models. Specifically, a combination of exponential piston-flow models (EPMs) and binary-mixing models (BMMs) (that comprise two EPMs) were used. BMMs were employed for wells where long time-series data are available for multiple tracers, and where an adequate fit to different tracer signals could not be obtained on the basis of a single EPM. Relative EPM mixing fractions were specified on the basis of aquifer confinement conditions and well-screen length (mixing fractions of 80-95% were

applied for wells with a long screen in unconfined conditions, whereas mixing fractions of 50-60% were applied for wells with shorter screens in confined conditions). The reader is referred to Morgenstern et al. (2018) for more details.

A highly parameterized approach was adopted (e.g., Hunt et al., 2007; Knowling et al., 2019), involving a total of 822 uncertain parameters. Spatially-distributed parameterization of hydraulic conductivity (horizontal and horizontal/vertical anisotropy ratio), effective porosity, specific storage and specific yield is achieved using pilot points (e.g., Doherty, 2003). Spatially-

distributed river-bed and boundary conductance parameters are defined on a reach and zone basis, respectively. We refer the reader to the Supplementary Information for more information.

### 2.4 Uncertainty quantification and data-worth exploration

Here we employ FOSM techniques (e.g., Tarantola, 2005; Doherty, 2015) to investigate the theoretical worth of various observation data types in terms of the their influence on the uncertainty variance surrounding forecasts following data assimilation.

Application of FOSM in this context requires only consideration of the relative differences in estimated forecast variance as a result of conditioning on different observation data types. Use of FOSM in relative contexts has been shown to be especially robust (e.g., Dausman et al., 2010; Herckenrath et al., 2011; Knowling et al., 2019).

The theoretical underpinnings of FOSM-based uncertainty quantification and data-worth assessment and details related to its application herein are presented in Appendix A.

Aspects that are relevant to the application of FOSM herein include:

- The prior parameter covariance matrix $\Sigma_{\theta}$ was specified as a block-diagonal matrix whereby geostatistical correlation between pilot-point based spatially-distributed parameters is represented through use of an exponential variogram with a range of approximately 10,000 m, and a sill proportional to the expected prior variance (the range of the square-root of the diagonal elements of $\Sigma_{\theta}$, i.e., the standard deviation of prior parameter uncertainty, is given in the Supplementary

Information). Non-spatially- and temporally-distributed parameters are assumed to be uncorrelated and therefore occupy diagonal matrix elements only.

- The Jacobian matrix $\mathbf{J}$ was populated using 1% two-point derivative increments.

– The diagonal elements of the epistemic noise covariance matrix $\Sigma_\epsilon$ (see Appendix A) was specified on the basis of observation "weights", adjusted in such a way that the measurement objective function equals the number of non-zero weighted observations, in order to approximate epistemic noise (i.e., the combined impact of random measurement errors and model simplification errors) based on model residuals (e.g., Doherty, 2015).

## 2.5 Results

For the summer spring discharge forecast in the central Heretaunga Plains, MRT observations display a worth that is considerably less than that of spring discharge observations during the summer months (i.e., when lower flows persist) and transient head observations (Figure 2 A). This is not surprising given that the forecast and the summer spring discharge observations are of the same type and represent the same temporal condition, and transient head observations are plentiful (5,704), spanning different time periods at annual, monthly and daily resolutions. The worth of MRT observations is greater than winter spring discharge observations, indicating a higher relevance of the spatially and temporally integrated information contained within MRT observations for this low-flow related prediction compared to the higher frequency and magnitude signals captured within spring discharge observations during winter.

Similar results from a relative perspective are apparent for the summer spring discharge forecast in the upper portion of the Heretaunga Plains. That is, transient head observations and spring discharge observations during summer are of highest worth, followed by observations of time-averaged heads, MRT and winter spring discharge (Figure 2 B)—for reasons described above. The greater worth of MRT observations for this forecast compared to the summer spring discharge forecast located down-gradient indicates that this forecast is more senstive to (uncertain) model parameters that are conditioned through assimilating MRT observations. This is due to the fact that the forecast is located where the aquifer is unconfined and receives rainfall and river recharge—these recharge rates are informed by MRT observations and have a large influence on the forecast.

For the winter spring discharge forecast, the worth of MRT observations is lower than that of other observations (Figure 2 C). This indicates a low relevance of the spatially and temporally integrated information contained in MRT observations with respect to a forecast concerning higher frequency and magnitude signals. This is also supported by the relatively low worth of the time-averaged head observations due to the temporally integrated nature of these quantities. As expected, a significantly greater worth of spring discharge observations during winter is evident for this forecast due to the unique and directly relevant information content associated with discharge observations that capture high-flow transience signals.

Across the three forecasts, a significantly larger worth is evident when MRT observations are added to the observation dataset compared to when MRT observations are removed from the observation dataset (red versus blue; Figure 2). This indicates that correlation occurs between the information contained within MRT observations and other observations. This is generally in contrast to the more unique information contained within spring discharge observations.

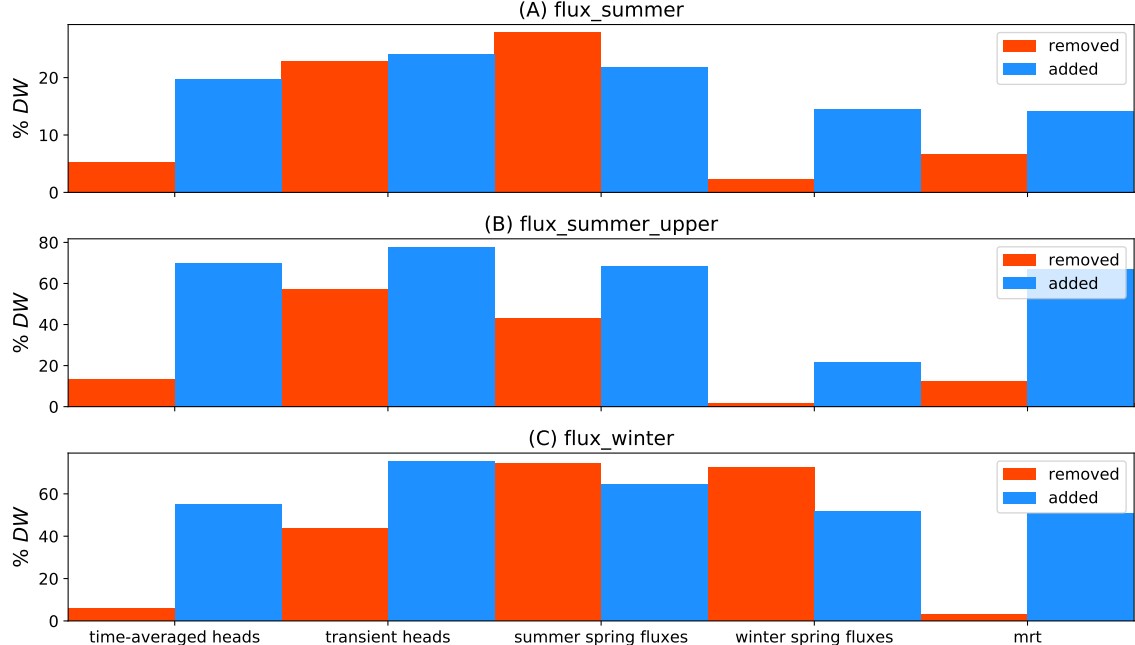

**Figure 2.** Worth of different observation groups (%DW) with respect to forecasts: (A) spring discharge flux during summer in central Heretaunga Plains; (B) spring discharge flux during summer in upper portion of Heretaunga Plains; and (C) spring discharge flux during winter in central Heretaunga Plains (see Figure 1 for locations). %DW is quantified as both the increase in forecast uncertainty variance following the removal of an observation group available for conditioning (red), and the decrease in forecast uncertainty variance following the addition of an observation group available for conditioning (blue) (see Appendix A). Note the different scales on the $y$-axes.

## 3    Second case study

The second case study serves to evaluate how assimilating discrete groundwater tritium concentration observations may affect
the robustness of forecasts in the context of a controlled model simplification experiment, where the simplification is related to
model vertical discretization (we refer the reader to White et al. (2020) for an exploration of the appropriateness of reduced-
discretization models in decision support more generally). In contrast to the first case study, which focused on the theoretical
worth of derived tritium observations in terms of changes in forecast variance, this case study proceeds with repeated data
assimilation in a paired simple/complex model analysis both with and without assimilating tritium observations. Through
these paired-model analyses, any potential biases or under-estimation of variances arising from the assimilation of tritium
observations with a simplified model can be exposed. A linked hydrologic-nutrient transport model of the Hauraki Plains (New
Zealand) (Figure 3) is used as a basis for the model simplification experiment.

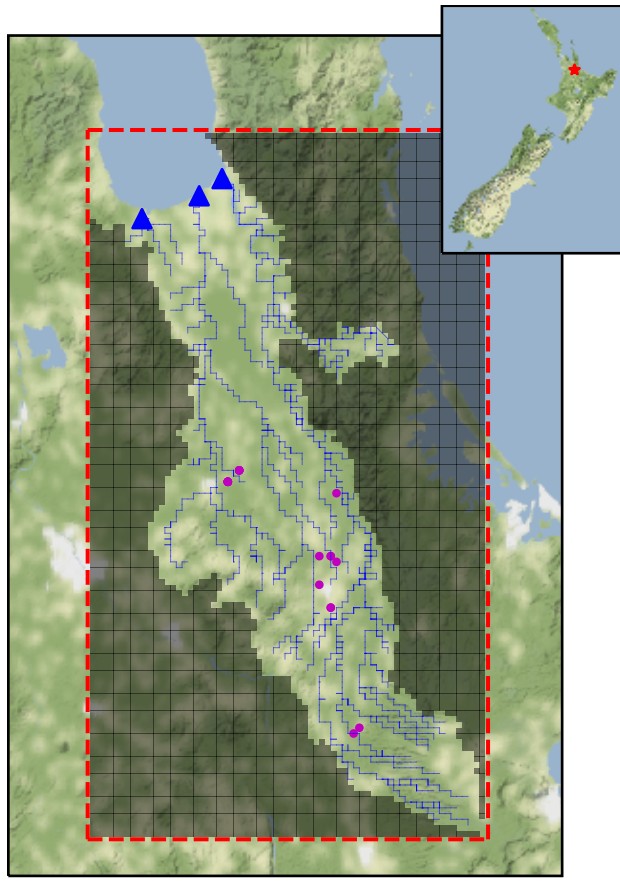

**Figure 3.** Hauraki Plains model extent (red dashed line), layer-1 inactive area (shaded), and surface-water network (blue lines). The terminal surface-water reaches that discharge to the Firth of Thames are shown as blue triangles. The location of groundwater tritium measurements are shown as magenta dots.

### 3.1 The model

The linked hydrologic-nutrient transport model simulates groundwater and surface-water flow using MODFLOW-NWT (Niswonger et al., 2011); advective and dispersive transport of nitrate and tritium in the groundwater and surface water system is simulated using MT3D-USGS (Bedekar et al., 2016). Denitrification and radioactive tritium decay processes are simulated using first-order reaction rates. The model is described in detail in White (2018), and the vertical-discretization simplification analysis is described in detail in White et al. (2020).

Herein, we focus on a single forecast: the cumulative load of nitrate discharging from the surface-water system to the Firth of Thames—an ecologically-sensitive estuary system—over a 10-year projection scenario involving present-day (2018) flow

and transport model forcing conditions. This forecast aggregates flow paths across the entire model domain (i.e., it represents the only nitrate-flux sink of the system). This forecast is referred to herein as the "Firth forecast".

## 3.2 Data assimilation and uncertainty quantification

As described in White et al. (2020), data assimilation was undertaken via history matching three versions of the model, each with a different vertical discretization scheme; history matching was performed using the iterative ensemble smoother PESTPP-IES (White, 2018).

History matching was conducted using 100 stochastic parameter realizations. An ensemble size of 100 was deemed sufficient to avoid under-utilization of observation data (i.e., "under-fitting") based on an exploration of the solution-space dimensionality using a subspace analysis (Moore and Doherty, 2005) (see the Supplementary Information and Knowling et al. (2019) for more details). Following history matching, the 10-year projection scenario was evaluated with the 100 history-matched realizations (effectively a 100-member sample of the posterior distribution). From the resulting 100 scenario evaluations, a posterior probability density function (PDF) of the First forecast was constructed.

The reader is referred to White (2018) and White et al. (2020) for a full description fo the Hauraki Plains model data assimilation process; a brief overview is nevertheless provided as follows:

– *Model parameterization.* Spatially-distributed parameterization of (horizontal and vertical) hydraulic conductivity, effective porosity, recharge rate, first-order denitrification rate, initial concentration and dispersivity is achieved using a combination of cell-based and zone-based multipliers. Nitrate-loading rate and abstraction well rate is parameterized using cell-by-cell and well-based mulitipliers, respectively. Streamflow-routing (SFR) elements are parameterized on a stream-segment basis. This parameterization approach gives rise to a problem dimensionality of 141268, 50180 and 29050 for the 7-layer, 2-layer and 1-layer model history-matching experiments, respectively. We refer the reader to White (2018) and White et al. (2020) for more information on parameterization and construction of prior parameter covariance matrices.

– *Observation data for assimilation.* The history-matching experiments included 20 tritium concentration observations from the groundwater system (Figure 3) (see also Supplementary Information for observation locations per model layer). Other observations such as long-term averaged groundwater levels and surface-water flows, and transient surface-water and groundwater nitrate conentrations were also used for history matching (see the Supplementary Information for observation locations).

As shown in White et al. (2020), the reduced-discretization (1-layer and 2-layer) model posterior PDFs for the Firth forecast display significant bias compared to the corresponding 7-layer model posterior PDF (Figure 4 A,D,G). In White et al. (2020), it was hypothesized that the tritium observations were giving rise to the apparent bias in the 1-layer and 2-layer posterior PDFs through the phenomenon of (inappropriate) parameter compensation (e.g., Clark and Vrugt, 2006; White et al., 2014) arising from history matching models with simplified model vertical discretization. Herein, we test this hypothesis by conditioning all three uniquely-discretized models again, but without using the discrete tritium observations, and then comparing the resulting

posterior PDFs to the corresponding PDFs in White et al. (2020). Any apparent difference in the posterior PDFs for the Firth
forecast is therefore directly attributable to the exclusion of the tritium observations during history matching.

## 3.3   Results

The process of history-matching with and without available groundwater tritium concentration observations yields substantial
differences in the posterior PDFs of the Firth forecast (Figure 4). In the case of the 7-layer "complex model" (Figure 4 A,B),
excluding the tritium observations results in a posterior PDF with a larger second moment and a slightly larger first moment
compared to including tritium observations for history matching; the difference between the Firth forecast posterior PDFs with
and without assimilating tritium observations is between 0 and $2 \times 10^7$ kg of nitrate (Figure 4 C). The larger second moment of
the posterior PDF when excluding tritium observations represents an intuitive and expected outcome: using fewer observations
for parameter conditioning through history matching should (theoretically) result in a larger posterior variance for the forecasts
that depend on those parameters.

Herein, for the purposes of identifying bias, the 7-layer model is considered to represent the best-available estimate of
the Firth forecast. Using this construct, we see that there are significant differences in posterior PDFs across the uniquely-
discretized models arising from data assimilation that included the tritium observations (Figure 4 A,D,G). This is largely in
contrast to the case where data assimilation is undertaken without the tritium observations, which leads to much more subtle
differences in posterior PDFs across the uniquely-discretized models (Figure 4 B,E,H).

The bias apparent in the posterior difference PDFs for the reduced-layer models relative to the 7-layer model (Figure 4 C,F,I)
are directly attributable to the use of tritium observations in the data assimilation process. The difference between the Firth
forecast PDFs resulting from data assimilation with and without tritium is most pronounced for the 1-layer model (Figure 4
I). In this case, excluding tritium observations from the history matching results in a decrease in simulated nitrate discharge
of $2 \times 10^7$ to $4 \times 10^7$ kg—approximately a 40% decrease in simulated mean nitrate discharge. We attribute the apparent 1-
layer PDF bias to the loss of simulated vertical flow and associated deeper groundwater flow paths. Briefly, this occurs due to
the aggregation of numerical discretization effects—the flow paths of a coarser-layer model will be a smoother and averaged
representation of those derived from a finer-layer model. While these deeper flow paths are not important for simulating the
nitrate transport cycle (given the relatively high denitrification rates in the Hauraki system), it is apparently important for
assimilating the tritium concentration observations.

The biases identified reflect the sensitivity of the Firth forecast to uncertain parameters that were conditioned by tritium
concentration observations. This occurs due to the spatially integrated nature of the Firth nitrate-load forecast, and because
the tritium observations provide insight into spatially and temporally averaged recharge and lateral flux rates in the upgradient
portion of the domain, where most of the surface-water/groundwater exchange occurs.

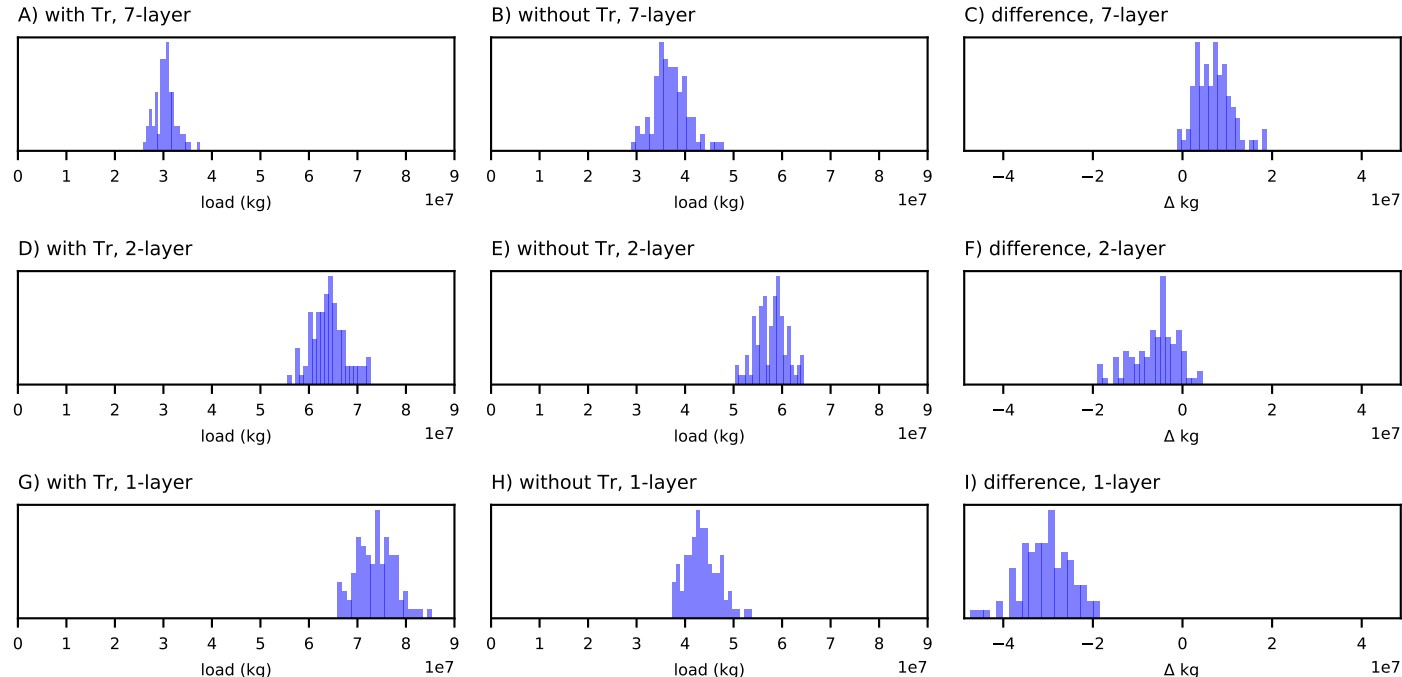

**Figure 4.** Comparison of posterior probability density functions (PDFs) for the Firth forecast. The left column (A,D,G) allows identification of bias as a result of both model simplification and tritium assimilation. By comparing to the middle column (B,E,H), model simplification-induced bias can be separated from that induced by assimilating tritium observations. The isolation of tritium assimilation-induced bias evident with different simplified models is shown in the right column (C,F,I). Including tritium observations in the conditioning of the 1-layer and 2-layer models (G,D) yields significant bias compared to the 7-layer PDF (A). However, if tritium observations are excluded from conditioning, the 1-layer and 2-layer PDFs (H,E) have considerably less bias compared to the corresponding 7-layer PDF (B). The differences in the PDFs (C,F,I) show that the tritium observations have the greatest biasing effect on the Firth forecast for the 1-layer model.

## 4 Discussion and Conclusions

This study explores the ramifications of assimilating tritium concentration and tritium-derived interpretation observations, specifically in the context of two examples of decision-support modeling. The benefit or otherwise of tritium data in other contexts such as site system characterization/understanding and conceptual model development is therefore not the focus of the current study—this study is concerned with a model's ability to "predict" (in two decision-support contexts) rather than "explain" (observed system behavior), as contrasted by Shmueli (2010).

The first case study presented herein serves to demonstrate that assimilating the rich information contained within tritium-derived MRT observations may be of variable worth in terms of improving the reliability of forecasts, especially where MRT observations are correlated with other available state observations (e.g., where hydraulic data are widespread, given the apparent spatially and temporally integrated information content of MRT observations, as supported by Ginn et al. (2009)). Moreover, the

worth of MRT observations is shown to vary between forecasts in such a way that reflects the underlying physics represented by the model (e.g., the MRT observations are of greatest worth for forecasts that are located where the aquifer is receiving recharge)—these physics dictate the "information flow" rather than the spatial proximity of the MRT observations and the forecast. The forecast-specific nature of observation worth has also been reported previously (e.g., Dausman et al., 2010; Fienen et al., 2010; White et al., 2016). The worth of MRT observations relative to various hydraulic potential and discharge observations across the different forecasts are, in general terms, similar to those reported by Hunt et al. (2006), Masbruch et al. (2014), Oehlmann et al. (2015) and Zell et al. (2018) (especially when considering the discussion point in the following paragraph).

While the particle-tracking model used in the first case study provides a mechanism for MRT observations to inform uncertain model parameters, including aquifer porosity (which is otherwise uninformed by other historical field observations), it is important to note that the forecasts are insensitive to porosity. That is, the information contained within MRT observations is spread between parameters that both do and do not play a role in constraining forecasts—effectively "diluting" the information available for conditioning. It is therefore expected that the worth of MRT observations presented herein would generally be larger for forecasts that are dependent on both uncertain hydraulic and transport parameters (e.g., particle travel times). This is notwithstanding that the uncertainty variance for such forecasts may be larger given the additional source of uncertainty associated with porosity. These findings are nevertheless highly relevant in that MRT observations are widely used and often regarded to be of benefit in constraining uncertain model parameters more generally (Schilling et al., 2019).

The second case study serves to demonstrate that assimilating tritium concentration observations with simplified (i.e., imperfect) numerical models may induce significant bias in forecasts—bias that is undetectable without a complex/simple model pair (e.g., Doherty and Christensen, 2011; White et al., 2014; Knowling et al., 2019). The forecast bias revealed in the second case study occurs as a result of the vertical discretization-simplified model's inability to appropriately assimilate the rich information content of the tritium observations. Generally, the observed pattern of simplification and resulting forecast bias implies that as the simplification of the model increases, the dangers of assimilating rich and diverse data types also grows. This result is highly relevant to decision-support modeling practitioners since all numerical models are gross simplifications of real environmental systems that they attempt to simulate. We refer the reader to Knowling et al. (2019) and White et al. (2020) for a broader exploration of the consequences of model simplification (in the form of parameterization reduction and vertical-discretization coarsening respectively) in terms of the decision-relevant forecast bias-variance trade-off and its implications for management decision making more generally.

Collectively, these results suggest that the assimilation of tritium and tritium-derived observations through history matching with an imperfect model should be strategic and approached with caution. It is recommended that these information-rich observations should not indiscriminately be incorporated in a data assimilation framework, given that this study has shown that such an approach ($i$) may be of variable apparent benefit, depending on the forecast being made, and ($ii$) when using imperfect models, may produce far worse forecast outcomes than those that would have been arrived at without assimilating these observations at all. This recommendation is similar to those by Brynjarsdóttir and O'Hagan (2014) and He et al. (2018). We consider this recommendation to be in stark contrast to what we believe is a common view among practitioners that "calibrating to more

data improves the model and its predictions"; we therefore consider this recommendation to be of significant implication to decision-support environmental modeling practitioners.

Furthermore, we expect the above-mentioned issues associated with imperfect-model data assimilation to be relevant and largely transferrable to the assimilation of other environmental tracers, other information-rich observations and diverse data types more generally. This is because we consider the primary barrier to appropriate assimilation of tritium observation data encountered in the second case study to be fundamental challenges associated with extracting appropriate information from spatially-discrete concentration observations when using upscaled or simplified representations of hydraulic properties within a regional-scale model that simulates tracer concentrations using the advection-dispersion equation (e.g., Zheng and Gorelick, 2003; Riva et al., 2008). To the extent that simulated outputs corresponding to observed tracer concentrations are sensitive to model details or parameters that are "missing" in a simplified model (e.g., White et al., 2014), parameter compensation will occur (e.g., Clark and Vrugt, 2006). To the extent that the forecast of management interest is dependent on these biased parameter estimates, the forecast will also become biased, potentially leading to resource mismanagement. The ubiquitous nature of model error and the challenges in appropriately accounting for differences in, e.g., representative spatial scales between field observations and model-derived quantities, suggests that the ill-effects identified in this study such as history matching-induced bias are not unique to the specifics of our study (e.g., consideration of tritium as a tracer). The similar findings and recommendations of Brynjarsdóttir and O'Hagan (2014) and He et al. (2018) in the statistics and petroleum reservoir disciplines, respectively, also supports the potential for the transferability in our findings and recommendations to data assimilation in other envrionmental modelling contexts.

If diverse and information-rich data such as tritium and MRT observations are available, and data assimilation through history matching is deemed necessary and/or appropriate, then a targeted modeling approach is needed that identifies which of these data are relevant to the forecast. This is critical to avoiding the ill-effects of model error in the context of decision support modeling (e.g., White et al., 2014; Knowling et al., 2019), as well as to avoid adding unnecessary complexity (through processes and parameters) needed to simulate the equivalent values of the diverse data for assimilation purposes, which may greatly increase the computational cost of the modeling analysis.

It should be noted, however, that even when the forecast is well "aligned" with observation data (i.e., the forecast is solution-space dependent), some degree of parameter compensation will inevitably occur—all models are gross simplifications and therefore model parameters do not perfectly represent real-world properties (e.g., Clark and Vrugt, 2006; White et al., 2014). However, if the data used for assimilation are commensurate with the forecasts, then the ill-effects of model error may be expected to be negligible (e.g., Doherty and Christensen, 2011; Watson et al., 2013).

The above findings and recommendations suggest that there is a significant need to identify better ways to assimilate diverse observation types including tracer concentration and tracer interpretation observations in numerical models for decision support. An enhanced ability to assimilate tracer observations, for example, will likely require increased model complexity (including advanced discretization, process representation and parameterization) to provide appropriate assimilation of rich and diverse data types that operate across a range of spatial and temporal scales commensurate with a given forecast.

However, an important and challenging compromise will be encountered: the need for enough model complexity to appropriately assimilate rich and diverse observations, while simultaneously ensuring that this level of complexity does not preclude the application of formal data assimilation and uncertainty quantification techniques due to the associated numerical instability and excessive run times. The navagation of this trade-off is central to effective and efficient decision-support modeling practice. In the meantime, tracer data model assimilation should involve processing or transforming of concentrations into quantities that may be more useful and may guard against ill-effects of history matching imperfect models (e.g., by integrating observations in space and time) (e.g., Rasa et al., 2013; Knowling et al., 2019; White et al., 2020).

## Appendix A: First-order second-moment (FOSM) methodology

This section provides a description of the FOSM approach used in the first case study to quantify uncertainty variance and assess data worth.

The posterior covariance matrix of uncertain model parameters $\overline{\Sigma}_{\theta}$ can be approximated using the Schur complement (Golub and Van Loan, 1996; Tarantola, 2005):

$$\overline{\Sigma}_{\theta} = \Sigma_{\theta} - \Sigma_{\theta} \mathbf{J}^T \left[ \mathbf{J} \Sigma_{\theta} \mathbf{J}^T + \Sigma_{\epsilon} \right]^{-1} \mathbf{J} \Sigma_{\theta} \tag{A1}$$

where $\Sigma_{\theta}$ is the prior parameter covariance matrix, which is specified based on expert knowledge pertaining to site system characteristics, $\Sigma_{\epsilon}$ is the epistemic observation noise covariance matrix (often assumed to have non-zero diagonal elements only), which includes the effects of model structural errors and measurement errors, and $\mathbf{J}$ is the Jacobian matrix of partial first derivatives (i.e., sensitivities) of simulated model outputs with respect to parameters. The Schur complement can be considered a linearized form of Bayes equation to estimate the second moment of the parameter and forecast posterior distribution (e.g., Goldstein and Wooff, 2007; Christensen and Doherty, 2008; Dausman et al., 2010).

Equation A1 assumes a linear relation between model parameters and simulated outputs (i.e., the sensitivities encapsulated within the $\mathbf{J}$ matrix is independent of the parameter values $\theta$). It also assumes that parameter and epistemic uncertainty distributions are Gaussian (i.e., normal).

While the posterior parameter and forecast uncertainty variances yielded by FOSM may only be approximate (depending on the validity of the linear assumption), the computational efficiency with which a large number of different number of conditioning "experiments" can be performed is unparalleled—these experiments facilitate rapid evaluation of the worth of different types of observations to reduce forecast variance. In addition, a number of studies have shown support for its usage especially in a relative second-moment sense (e.g., Dausman et al., 2010; Herckenrath et al., 2011; Knowling et al., 2019).

The prior and posterior uncertainty variance surrounding a forecast $\sigma_s^2$ can be expressed by mapping uncertainty from parameter to forecast "space". This is achieved by computing the sensitivity of the forecast to model parameters, comprising the vector $\mathbf{y}$ (i.e., a row of $\mathbf{J}$). That is:

$$\sigma_s^2 = \mathbf{y}^T \boldsymbol{\Sigma}_{\boldsymbol{\theta}} \mathbf{y} \tag{A2}$$

and

$$\overline{\sigma}_s^2 = \mathbf{y}^T \overline{\boldsymbol{\Sigma}}_{\boldsymbol{\theta}} \mathbf{y} \tag{A3}$$

The worth of data, expressed as a percentage, is given by:

$$\%DW = \frac{|\sigma_{\pm obs}^2 - \sigma_{base}^2|}{min\{\sigma_{base}^2, \sigma_{\pm obs}^2\}} \times 100 \tag{A4}$$

where $\sigma_{\pm obs}^2$ is the increase/decrease in forecast uncertainty variance as a result of the removal/addition of one or more observations or observation groups used for parameter conditioning, respectively, and $\sigma_{base}^2$ is either the forecast uncertainty calculated on the basis of all observation data/zero observation data, depending on whether data worth is being quantified by adding or removing observations.

Herein, we quantify %DW as a result of both the removal and addition of observation groups. We primarily focus on %DW values based on the removal of an observation group from an otherwise full observation dataset available for assimilation, given that these values reflect the unique (i.e., uncorrelated) information content of observations. However, the difference between %DW values arising from these different data-worth quantification approaches is used herein to comment on the level of information uniqueness/redundancy within observation groups.

It is important to note that each FOSM-based data worth assessment is conducted with respect to a single forecast (notwithstanding that we evaluate the worth of different observation data with respect to a number of different forecasts). We consider this to be a side-benefit of this approach, especially given the need for decision-support modeling to be undertaken in a forecast-targeted manner, as discussed recently by White (2017).

*Author contributions.* MJK, JTW and CRM contributed to the concept. MJK and JTW undertook the modeling analyses. MJK prepared the manuscript with input from JTW. JTW and CRM contributed to the manuscript preparation. PR and KH contributed to the underlying Heretaunga Plains models.

*Competing interests.* The authors declare that they have no conflict of interest.

*Acknowledgements.* This research was performed as part of both the Te Whakaheke o te Wai and Smart Models for Aquifer Management Programmes, funded by the Ministry of Business, Innovation and Employment (New Zealand), with co-funding from Hawke's Bay Regional Council and Waikato Regional Council. The authors wish to thank Ty Ferre, Chris Turnadge and the anonymous reviewer for their helpful comments.

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

Pollock, D. W.: User guide for MODPATH version 6—A particle-tracking model for MODFLOW: U.S. Geological Survey Techniques and
Methods, U.S. Dept. of the Interior, U.S. Geological Survey Reston, Va, version 6. edn., 2012.

Rajanayaka, C. and Fisk, L.: IRRIGATION WATER DEMAND & LAND SURFACE RECHARGE ASSESS-
MENT FOR HERETAUNGA PLAINS, https://www.hbrc.govt.nz/assets/Document-Library/Publications-Database/
Aqualinc-Irrigation-water-demand-land-surface-recharge-assessment-Heretaunga-Plains-20180713.pdf, 2018.

Rakowski, P. and Knowling, M.: Heretaunga aquifer system groundwater model development report, https://www.hbrc.govt.nz/assets/
Document-Library/Publications-Database/4997-Heretaunga-Model-Groundwater-Development-Report.pdf, 2018.

Rasa, E., Foglia, L., Mackay, D. M., and Scow, K. M.: Effect of different transport observations on inverse modeling results: case study of
a long-term groundwater tracer test monitored at high resolution, Hydrogeology Journal, 21, 1539–1554, https://doi.org/10.1007/s10040-
013-1026-8, https://doi.org/10.1007/s10040-013-1026-8, 2013.

Riva, M., Guadagnini, A., Fernandez-Garcia, D., Sanchez-Vila, X., and Ptak, T.: Relative importance of geostatistical and trans-
port models in describing heavily tailed breakthrough curves at the Lauswiesen site, Journal of Contaminant Hydrology, 101, 1 –

13, https://doi.org/https://doi.org/10.1016/j.jconhyd.2008.07.004, http://www.sciencedirect.com/science/article/pii/S016977220800106X, 2008.

Sanford, W.: Calibration of models using groundwater age, Hydrogeology Journal, 19, 13–16, 2011.

Sanford, W. E., Plummer, L. N., McAda, D. P., Bexfield, L. M., and Anderholm, S. K.: Hydrochemical tracers in the middle Rio Grande Basin, USA: 2. Calibration of a groundwater-flow model, Hydrogeology Journal, 12, 389–407, https://doi.org/10.1007/s10040-004-0326-4, https://doi.org/10.1007/s10040-004-0326-4, 2004.

Schilling, O. S., Cook, P. G., and Brunner, P.: Beyond Classical Observations in Hydrogeology: The Advantages of Including Exchange Flux, Temperature, Tracer Concentration, Residence Time, and Soil Moisture Observations in Groundwater Model Calibration, Reviews of Geo-
525 physics, 57, 146–182, https://doi.org/10.1029/2018RG000619, https://agupubs.onlinelibrary.wiley.com/doi/abs/10.1029/2018RG000619, 2019.

Shmueli, G.: To Explain or to Predict?, Statist. Sci., 25, 289–310, https://doi.org/10.1214/10-STS330, https://doi.org/10.1214/10-STS330, 2010.

Siade, A., Prommer, H., Suckow, A., and Raiber, M.: Using Numerical Groundwater Modelling to Constrain Flow Rates and Flow Paths in
the Surat Basin through Environmental Tracer Data, https://doi.org/10.25919/5b8055bfe3ea5, 2018.

Stewart, M. K. and Thomas, J. T.: A conceptual model of flow to the Waikoropupu Springs, NW Nelson, New Zealand, based on hydrometric and tracer 18 0, Cl, 3 H and CFC) evidence, Hydrology and Earth System Sciences, 12, 1–19, https://doi.org/10.5194/hess-12-1-2008, 2008.

Tarantola, A.: Inverse problem theory and methods for model parameter estimation, SIAM, 2005.

Turnadge, C. and Smerdon, B. D.: A review of methods for modelling environmental tracers in groundwater: advantages of tracer concentration simulation, Journal of Hydrology, 519, 3674–3689, 2014.

Watson, T. A., Doherty, J. E., and Christensen, S.: Parameter and predictive outcomes of model simplification, Water Resources Research, https://doi.org/10.1002/wrcr.20145, http://dx.doi.org/10.1002/wrcr.20145, 2013.

White, J. T.: Forecast First: An Argument for Groundwater Modeling in Reverse, Groundwater, 55, 660–664,
https://doi.org/10.1111/gwat.12558, http://dx.doi.org/10.1111/gwat.12558, 2017.

White, J. T.: A model-independent iterative ensemble smoother for efficient history-matching and uncertainty quantification in very high dimensions, Environmental Modelling & Software, https://doi.org/https://doi.org/10.1016/j.envsoft.2018.06.009, http://www.sciencedirect.com/science/article/pii/S1364815218302676, 2018.

White, J. T., Doherty, J. E., and Hughes, J. D.: Quantifying the predictive consequences of model error with linear subspace analysis, Water
Resources Research, 50, 1152–1173, https://doi.org/10.1002/2013WR014767, http://dx.doi.org/10.1002/2013WR014767, 2014.

White, J. T., Fienen, M. N., and Doherty, J. E.: A python framework for environmental model uncertainty analysis, Environmental Modelling and Software, 85, 217 – 228, https://doi.org/http://dx.doi.org/10.1016/j.envsoft.2016.08.017, 2016.

White, J. T., Knowling, M. J., and Moore, C. R.: Consequences of model simplification in risk-based decision making: An analysis of groundwater-model vertical discretization, Groundwater, https://doi.org/10.1111/gwat.12957, http://dx.doi.org/10.1111/gwat.12957,
2020.

Wilding, T. K.: Heretaunga Springs: Gains and losses of stream flow to groundwater on the Heretaunga Plains, 2017.

Zell, W. O., Culver, T. B., and Sanford, W. E.: Prediction uncertainty and data worth assessment for groundwater transport times in an agricultural catchment, Journal of Hydrology, 561, 1019 – 1036, https://doi.org/https://doi.org/10.1016/j.jhydrol.2018.02.006, 2018.

Zheng, C. and Gorelick, S. M.: Analysis of Solute Transport in Flow Fields Influenced by Preferential Flowpaths at the Decimeter
Scale, Groundwater, 41, 142–155, https://doi.org/10.1111/j.1745-6584.2003.tb02578.x, https://onlinelibrary.wiley.com/doi/abs/10.1111/
j.1745-6584.2003.tb02578.x, 2003.

Zuber, A., Różański, K., Kania, J., and Purtschert, R.: On some methodological problems in the use of environmental tracers to estimate
hydrogeologic parameters and to calibrate flow and transport models, Hydrogeology journal, 19, 53–69, 2011.