# Peer review of "On the assimilation of environmental tracer observations for model-based decision support"

_Hydrology and Earth System Sciences, 2019_

## Short Comment (SC1) · 20 Sep 2019

This is another excellent paper from this group. To me, it sits right between academic and applied hydrology. The group has such advanced modeling skills, that they have immediate credibility when they point out limitations based on model-based interpretations. In this paper, they carry their analysis through to the 'value' of data for decision support. They make a great case that data, even if it inherently carries important information, can be misleading if it is viewed through the lens of an imperfect model. Of course, they are all imperfect models. This is really important work and I feel that HESS is just the right target audience. I hope that it inspires continued careful examination of

the interaction of data and models for decision support.

Well done Knowling et al!

Ty Ferre
* * *

---

## Short Comment (SC2) · 22 Sep 2019

We thank Prof. Ty Ferre for his positive and encouraging comments! We acknowledge his significant contribution to the field of decision-support modeling (the field in which this paper is cast).

His comment that our study strikes a balance between academic and applied hydrology is pleasing as it reflects our endeavor to tackle problems that are relevant to both researchers and practitioners.

His comments reiterate that our findings regarding the potential for assimilation of

information-rich tracer data to cause ill-effects can be extended beyond just the (imperfect model) tracer data assimilation context. This is covered in the Discussion and Conclusions section.

Thanks again to Prof. Ferre!

Matt Knowling

---

## Referee Comment (RC1) · Ty P. A. Ferre (Referee) · 4 Oct 2019

I provided an informal review previously, this serves as a somewhat more detailed formal review:

The authors continue to tackle one of the most important areas of applied hydrogeology with novel and insightful tools. Here, they challenge an accepted fact of hydrogeology – that more data, and especially more diverse data – will lead to better models for decision support. Their counterintuitive finding that isotopic data may have value little of no value for water resources is just the sort of result that could spark important conversations in our field.

[Figure]

As with their previous work, this group takes full advantage of their position as leading hydrogeologic modelers to offer constructive criticism for the field. There is no question that this group can build and calibrate large, complex models and that they are as able as anyone to extract information from hydrologic observations. This lends weight to what could otherwise be criticized as a finding based on lack of response. In this case, the group makes the point that increased complexity has a place in assimilating more and more varied data while recognizing that this increased complexity has limits for some applications. Again, the group takes advantage of its abilities to provide useful guidance for the community.

Personally, I appreciate the terse format of presenting the case studies. There may be call for providing more detail as supplemental information. I believe that the accepted White paper supplies these details. But, I will leave that to other reviewers and the editor to comment on whether it is appropriate to provide more detail in this manuscript.

My only recommendation with regard to the authors approach reflects my own bias. As such, I completely understand if they do not address it in the paper! Regardless, I would like to hear the authors respond to the following question. From the perspective of a water manager or someone else tasked with assessing hydrologic risk, is a statistical reduction in the forecast the right measure of data value? Would the value of tritium, in this case, be viewed differently if decision-making were seen to be based on hypothesis testing of the plausibility of future high loading, for example? More generally, could the authors comment on the importance of considering the decision making context underlying the assessment of data worth?

Fantastic work – I look forward to reading more in the series!

Ty Ferre

---

## Short Comment (SC3) · 11 Oct 2019

I would like to thank the authors for their invitation to provide comments on this manuscript. I believe this manuscript provides a valuable and timely contribution towards guidance in the use of subsurface environmental tracers to improve the predictive capability of groundwater flow and transport models. Many publications have implored researchers and practitioners to include a range of non-standard observation types such as temporal differences (Peeters et al., 2011), temperatures (Anderson, 2005), isotope concentrations and activities and inferred residence time (or "ages") (Schilling et al., 2019), and/or geophysical data (Hinnell et al., 2010) in model inversion

and prediction uncertainty minimisation. However, investigations of when benefits may be obtained (or perhaps more importantly, when not) from these additional data types (and therefore specific guidance in their use) has been limited.

On first reading of the manuscript I questioned whether the authors were "over-reaching" in their conclusions. Having re-read the manuscript, I now believe that the authors have been careful to state that their conclusions regarding the applicability of environmental tracer observations are indicative, rather than comprehensive. More generally though, I do believe the manuscript will benefit from some revisions. Specifically, I would like to provide three major criticisms of the manuscript. These mainly relate to the suitability of the experimental design, in terms of the suitability of testing the hypotheses presented. These are followed by a number of minor criticisms that I believe the authors should also consider addressing. These minor criticisms mostly relate to the interpretation of environmental tracers, or to descriptions provided of the parameterisations of the numerical models used.

I hope my comments are helpful to the authors in improving the manuscript and I am more than willing to provide further clarifications off-line, if they are needed.

Major comments:

1. Tracer observations are of limited additional value when direct observations of fluxes are already available

For the Heretaunga Plains example, I do not believe that the addition of environmental tracer observations (in this case, tritium) when flux observations (in this case, spring discharge) are already available provides an ideal (i.e. fair) test case. Tracer concentrations are proxies for fluxes (either recharge, lateral or discharge) so they are often measured (and subsequently included in model inversion) when direct observations of fluxes are not available. Assessing the value of tracer observations when flux observations are not available would provide a fairer test case and would be of greater interest. For the Heretaunga Plains example, this could be implemented by simply

omitting spring discharge observations from all model inversion.  

2. Tracer observations are of limited additional value when collected upstream of prediction locations

For the Hauraki Plains example, subsurface tritium observations were recorded far upstream of nitrate discharge predictions. Since environmental tracers such as tritium integrate information along flow paths, it is intuitive that these tritium observations would contain limited information of value to predictions of nitrate concentration located far downstream. If subsurface tritium concentrations are available from locations in the vicinity of the predictions of interest, then I suggest that it would be more relevant to include these in the case study.

3. Conservativeness and management thresholds when presenting prediction uncertainty

For the Hauraki Plains example, I believe that the authors criticise simple modelling approaches without providing any discussion of whether modelled predictions are conservative. Prediction histograms are presented in the absence of a management threshold; in this case, an upper permissible limit for nitrate discharge. I suggest that the authors present a relevant management threshold when presenting prediction uncertainty results. This would allow the authors the additional benefit of exploring whether predictions produced using simple and complex parameterisation approaches were conservative.

Minor comments

1. Mean residence times

Mean residence time (MRTs) values were used to quantify the age of groundwaters. However, MRTs are equivalent to groundwater ages only under very strict assumptions; specifically, requiring highly simplified conceptualisations. The latter are the basis of mixing functions used in lumped parameter models. In most (complex, real-world)

[Figure]

cases, MRTs act as a fitting parameter. This is especially true if MRTs are derived from binary mixing models, which arbitrarily blend two mixing models, which generally undermines their physical bases.

As a solution, and rather than the use of lumped parameter models to derived mean residence times, where possible I suggest simulating the reactive transport (or at least, non-reactive transport with first order decay) of environmental tracer concentrations (or activities. Admittedly, for some tracers (such as carbon-14) the complexity of reactions may make reactive transport simulation prohibitive. However, the examples presented by the authors already feature combined numerical flow and transport models. Additionally, the authors' examples also feature a relatively simple tracer requiring only the simulation of first order decay, so I would assume that reactive transport simulation would be feasible.

2. Binary mixing models

The authors state that, as part of lumped parameter modelling, binary mixing models (BMMs) were used to derive mean residence times from subsurface tritium concentrations. BMMs are a linear combination of two other (ideally) physically-based mixing models. I suggest that the authors describe which mixing models were combined using the BMMs, the relative contribution of each, and the physical meaning of the combined result.

3. Atmospheric tritium concentrations

The authors state that the historic record of atmospheric tritium concentration features a "shape [that allows] for unique interpretation" (lines 69–70). It is unclear what the authors mean by the term "unique" in this context. Unlike SF6, it would not be true to state that the historical atmospheric tritium record is consistently monotonic. Tritium values increased initially due to nuclear weapons testing and have declined ever since. In some cases, the historical atmospheric tritium record features more than one peak value after the cessation of nuclear weapons testing. In addition, historic records of

atmospheric tritium concentrations at various locations are typically quite noisy, unlike SF6. See Figure 1 in McCallum et al. (2014) for an example of a noisy, two-peak example of a historical atmospheric tritium record. For this example, and following correction for radiometric decay in the subsurface, a tritium observation of 20 TU would correspond to any of four different recharge times (and therefore ages). In comparison, a historical atmospheric SF6 record is also shown by McCallum et al. (2014), which increases monotonically and would therefore permit unique interpretations of "ages" from measured concentrations. I suggest that the authors could state instead that tritium is a popular tracer for the identification of young age groundwaters (i.e. <70 years old) for the following reasons. Unlike CFCs, it is not affected by microbial degradation or contamination and, unlike SF6, it is not affected by potential subsurface sources. The authors may wish to cite Beyer et al. (2014), who provided a comparison of traditional (e.g. 3H, CFCs, SF6) and emerging (e.g. Halon-1301, SF5CF3) young age tracers.

4. Non-reactive modelling of contaminant transport

The Hauraki Plains model simulated nitrate as a non-reactive constituent. In practice, nitrates in the subsurface are subject to a range of processes: assimilation, nitrification/denitrification, volatilisation, sorption/desorption and retardation (Kendall and Aravena, 2000). For this reason, I suggest that the authors explain why nitrate was not simulated as a reactive constituent in the forward model.

5. Screen lengths of wells sampled for environmental tracers

When interpreting subsurface concentrations of environmental tracers, knowledge of the length of screened sections in sampled groundwater wells is crucial. If lumped parameter models are used, this affects the choice of mixing model. For example, if sampled wells are open holes or fully screened then the exponential mixing, exponential–piston flow, or dispersion models may be appropriate. Alternatively, if sampled wells are partially screened then the partial exponential model may be appropriate. Given the importance of this information to tracer interpretation, I suggest that the authors

describe the screen extents of each sampled well. The authors could also state how this information was used to select an appropriate mixing model for lumped parameter modelling, from which mean residence times were calculated.

6. Pilot point parameterisation

Pilot point parameterisations of the Heretaunga Plains and Hauraki Plains models are not described explicitly in the manuscript. Specifically, it is not clear which parameters were parameterised using this method. I suggest that the authors describe explicitly which model parameters were implemented on a cell-by-cell basis, or using pilot points, zonation or using spatially uniform values, including horizontal and vertical hydraulic conductivity, specific yield/storage, recharge and, for transport models, porosity.

7. Variogram definitions

It is not clear whether, for a given model, the same variogram was used to implement pilot point parameterisation for one or many parameter types. For example, I would not expect that the spatial correlation between hydraulic conductivity values to be the same as for recharge rates. I suggest that the authors state explicitly which variogram parameter values (e.g. correlation length, range, sill, nugget) were used to define which model parameter values, and describe the spatial analyses used to quantify spatial correlation between parameter values. Given that the degree of model complexity (particularly in relation to the ability of a model to assimilate observed data) is a key focus of the manuscript, I believe that detailed descriptions of the model parameterisation used are relevant.

8. Bias and underestimation

The authors state that the assimilation of tritium can induce "biased first moments or underestimated second moments" (line 55). I suggest that the authors could unpack this statement by providing simple examples to support this statement, for both bias and underestimation. The authors could also state explicitly the nature of the bias; i.e.

[Figure]

whether prediction mean values were under- or overestimated.

9. Ensemble size representativeness

The authors state that their implementation of the Iterative Ensemble Smoother featured ensemble sizes of 100 (lines 194-195). This value appears to have been selected arbitrarily, likely based on logistical constraints (e.g. forward model and inversion computing times). Was bootstrapping or other representativeness/convergence testing methods used to assess whether an ensemble size of 100 representative, and/or whether ensemble statistics converged as the ensemble size approached 100? I suggest that the authors demonstrate that the ensemble size used was representative.

10. Vertical coarsening of model grid

The authors provide limited explanation of why vertical coarsening of the model grid led to fewer relatively long flow paths. Since this observation is crucial to the interpretation of the authors' results, I suggest that the authors expand their discussion of this key point.

References cited:

Anderson, M. P. (2005). Heat as a ground water tracer. Groundwater, 43(6), 951-968.

Beyer, M., Morgenstern, U., and Jackson, B. (2014). Review of techniques for dating young groundwater (< 100 years) in New Zealand. Journal of Hydrology (New Zealand), 53(2), 93-111.

Hinnell, A. C., Ferré, T. P. A., Vrugt, J. A., Huisman, J. A., Moysey, S., Rings, J., and Kowalsky, M. B. (2010). Improved extraction of hydrologic information from geophysical data through coupled hydrogeophysical inversion. Water Resources Research, 46(4).

Kendall, C., and Aravena, R. (2000). Nitrate isotopes in groundwater systems. In: Environmental Tracers in Subsurface Hydrology. Cook, P. G., and Herczeg, A. L. (Eds.). Springer, Boston, Massachusetts, USA, pp. 261-297.

McCallum, J. L., Engdahl, N. B., Ginn, T. R., and Cook, P. G. (2014). Nonparametric estimation of groundwater residence time distributions: What can environmental tracer data tell us about groundwater residence time?. Water Resources Research, 50(3), 2022-2038.

Peeters, L. J. M., Rassam, D., and Lerat, J. (2011). Improving parameter estimation in transient groundwater models through temporal differencing. In: Chan, F., Marinova, D., Anderssen, R.S. (Eds.), MODSIM2011, 19th International Congress of Modelling and Simulation, Modelling and Simulation Society of Australia and New Zealand; Perth, WA, Australia; 12–16 December 2011, pp. 3959-3965.

Schilling, O. S., Cook, P. G., and Brunner, P. (2019). Beyond Classical Observations in Hydrogeology: The Advantages of Including Exchange Flux, Temperature, Tracer Concentration, Residence Time, and Soil Moisture Observations in Groundwater Model Calibration. Reviews of Geophysics, 57(1), 146-182.

---

## Short Comment (SC4) · 15 Oct 2019

We thank Mr Chris Turnadge for his detailed and constructive comments. We feel that his comments can be addressed where appropriate through some minor yet important additions and modifications to the manuscript text. His comments also provide us with an opportunity to reiterate and expand on some of our current decision-support modeling perspectives and philosophies. We respond to each of his comments below.

We agree that the guidance provided in the manuscript is timely for the reasons Mr Turnadge describes (and for reasons described in the Introduction and Discussion sections of the manuscript). It was one of our primary motivations to rigorously investigate

the calls for increased use of diverse data (e.g., Schilling et al., 2019) in the context of decision-support modeling. We thank Mr Turnadge for his positive sentiments.

We agree that our conclusions are "indicative", and we are pleased Mr Turnadge believes our carefully formulated conclusions are appropriate. We feel that indicative conclusions are really all that can be drawn on the basis of two (or even more) real-world case study example demonstrations. We also consider empirical demonstrations to provide an important adjunct to theoretical demonstrations; we feel empiricism in the presence of inevitable site specifics are important to accompany theoretical investigation.

Response to major comments:

1. First, our results do in fact already show the worth of MRT observations in the absence of spring discharge observations (albeit in the absence of other observations too—therefore representing the case where the maximum worth of MRT observations is apparent with an otherwise "empty" observation dataset). Please see the blue bars in the MRT column of Figure 2. Second, the Heretaunga Plains case study presented reflects a real-world investigation, i.e., whereby tritium concentrations were measured after and in combination with discharge measurements. It is not the experience of the authors that tracer concentrations are typically only sampled where flux observations are lacking. This would imply that flux and tracer data contain the same information and can therefore be substituted for one another. Contrast this with much literature suggesting the benefits of as many data types and as much data as possible. For these reasons, we consider the first case study to represent a fair and useful test case.

2. We regret that this comment appears to reflect a misunderstanding (and therefore a need to improve the communication of the manuscript) more than anything else. While we agree with Mr Turnadge's intuition regarding the integration of tracer-derived information along flow paths, we do not show that tritium concentration observations in the second case study are of limited value or "worth" (regardless of where the observations
occurred within the basin). In fact, despite that the second case study does not actually explore the worth or information content of tritium observations, the results suggest that the tritium observations contain "too much" information (or, rather, misinformation when considered through the lens of an imperfect model that lacks parameter receptacles for the information contained in the tritium observations). That is, the sensitivity of the forecast of interest to uncertain parameters that were conditioned by tritium concentration observation led to forecast bias. This occurs due to two factors: 0. the "Firth" nitrate-load forecast aggregates flow paths across the entire domain (i.e., this forecast represents the only nitrate flow sink of the system) and in time; and 1. the tritium observations provide insight into spatially and temporally averaged recharge and lateral flux rates in the upgradient portion of the domain, where most of the surface-water/groundwater exchange occurs. In other words, the bias reflects the information content of the upgradient tritium observations related to averaged upgradient model parameters on which the forecast is sensitive. To address this comment, we will add text regarding the nature of the forecast to the Second case study section, and this explanation to the Discussion and Conclusions section.

3. We agree that exploration of the conservativeness or otherwise of simplified model forecast PDFs (the former of which can be viewed as a metric for accepting such a simplified model) is an important undertaking, especially when performed with respect to a specific management decision threshold. We are pleased to inform Mr Turnadge that we have two papers that explicitly tackle this question—the first in terms of model parameterization (Knowling et al., 2019) and the second in terms of model vertical discretization (White et al., forthcoming; we will send this manuscript to Mr Turnadge). We will therefore address this comment by adding an explicit reference to these manuscripts in the Discussion and Conclusions section. We also refer Mr Turnadge to Prof. Ferre's related question in his review of the manuscript regarding how consideration of a management decision threshold may provide a more appropriate basis for assessing data worth from a decision maker's perspective; we will respond to this comment shortly.

Response to minor comments:

1. We agree with Mr Turnadge regarding the simplicity and lack of physical basis of LPMs, and the potential benefits of full advective-dispersive (and reactive) transport numerical models for simulating tracer concentrations (as described comprehensively by Turnadge and Smerdon, 2014), notwithstanding their practical limitations, which are amplified in formal decision-support modeling contexts. Importantly, we reiterate here that LPM-derived MRT observations are only used (in combination with advective-transport simulations) in the first case study; the second study employs full advective-dispersive transport modelling together with a first-order reaction rate to simulate radioactive decay of tritium. Our intention here was to employ "standard practice" tracer modeling techniques as a basis for exploring the ramifications of model tracer-data assimilation, such that the findings are as useful as possible to industry. The literature reflects the common use of both advective-only particle-tracking simulations (combined with LPM-based "age" observations) and advective-dispersive simulations (combined with tracer concentrations) (e.g., Gusyev et al., 2014). We will address this comment by presenting the above explanation and justification in the revised manuscript, and also by making explicit mention of reactive-transport modeling approaches as one means of increased model complexity that may facilitate improved imperfect model-data assimilation.

2. The MRT observations in the first case study were derived using a combination of exponential piston flow models (EPMs) and BMMs (comprising two "parallel" EPMs), as described in detail in Morgenstern et al. (2018). For the EPMs, Morgenstern et al. (2018) states "For wells with a long well screen interval in unconfined conditions, a high fraction of exponential (mixed) flow of 80–95% was applied. For wells with a narrow screen interval in confined conditions, a low fraction of exponential flow of 50–60% was used". BMMs were employed for most of the drinking-water wells where long time-series data are available for multiple tracers and where an adequate LPM fit (to different tracer signals) could not be obtained on the basis of a single EPM (e.g., due to

complex geological features). We will add these details to the First case study section of the manuscript.

3. We agree with the need to be more specific regarding atmospheric tritium concentrations. We thank Mr Turnadge for his suggested revision to the text. We will revise the manuscript directly following his suggestion.

4. The second case study does in fact use a first-order decay rate to simulate the process of denitrification reactions. We consider this approach to be "standard" modeling practice. We address this comment by making this point clear in the Second case study section of the revised manuscript. Note that we do not simulate the full range of nitrate reactive processes explicitly simply due to computational resource constraints; these constraints become more limiting where model deployment is undertaken stochastically for decision-support purposes.

5. We agree with Mr Turnadge regarding the importance of the well screen length for tracer interpretation—especially when using LPMs for interpretation. As described in response to minor comment (2), we will indicate the role that well screen lengths had on the deployment of LPMs to infer MRT in the First case study section.

6. We will add a description to both case study sections of the revised manuscript listing the parameterization device (e.g., spatially uniform, zones, pilot points, grid-based) employed for each parameter type. Briefly, for the Heretaunga Plains case study, pilot points were used to parameterize hydraulic conductivity (horizontal and horizontal-vertical anisotropy ratio), effective porosity, specific storage and specific yield, while river-bed and boundary conductance parameters are defined on a reach and zone basis. For the Hauraki Plains case study, the parameterization approach has already been described in detail in White (2018) and Knowling et al. (2019); we will therefore add only a brief summary to the revised manuscript along with a reference to these papers.

7. We will address this comment through indicating which spatially distributed parameters are represented by which variograms. Briefly, for the Heretaunga Plains case study, the same variogram (variogram parameters already defined in manuscript) is used for pilot-point based distributed parameters (only subsurface property-related parameters); no spatial correlation is assumed otherwise. For the Hauraki Plains case study, variogram details regarding the various different parameter types in the second case study have already been described in White (2018) and Knowling et al. (2019); we will therefore add only a brief summary to the revised manuscript along with a reference to these papers. On a more general note, we agree that variograms could theoretically be defined on a parameter type-specific basis from a physically-based (or perhaps more appropriately a "physically-motivated") parameter standpoint. However, given our recent experience and findings regarding the significant potential for ill-effects (e.g., forecast bias) in real-world decision-support modeling (e.g., Knowling et al., 2019; White et al., forthcoming), we tend to consider spatially distributed parameters employed by regional-scale models to be significant "abstractions" (i.e., from their intended property representation; e.g., Watson et al., 2013). It follows that questions such as "what is the variogram for spatially distributed recharge bias-correction parameters?" and "how do we represent uncertainty in the variogram model used to describe prior parameter correlation and heterogeneity (i.e., a "hyper-parameter")?" arise when trying to rigorously deal with real-world model error.

8. On Line 55, we are making only a general statement that data assimilation through history matching an imperfect model can result in forecast bias and uncertainty underestimation; these ill effects occur as a result of both model simplification and history matching. We cite literature that demonstrate these phenomena. This statement does not relate to tracers or tritium specifically, or any other data in particular. Therefore, no comments can be made at this point as to the nature of bias or variance underestimation. The "direction" of tritium assimilation-induced bias (i.e., under- or over-estimation) and is covered in the Results section of the second case study (although we note that the direction of bias may not be very generalizable between different forecasts and between different sites). Nevertheless, we will revise this sentence to unpack this

sentence by defining the terms "first moment" and "second moment" explicitly here.

9. The ensemble size was in fact selected on the basis of an approximation of the solution space dimensionality. This approximation was obtained through a subspace analysis of predictive error variance (Moore and Doherty, 2005). We refer Mr Turnadge to the Supplementary Material of Knowling et al. (2019) for more information on this, including a plot of the singular value spectrum.

10. Fewer relatively long flow paths occur when vertically coarsening the model grid simply due to the aggregation of numerical discretization effects—the flow paths of a coarser-layer model will be a smoother and averaged representation of those derived from a finer-layer model. We agree that this is an important explanation to support the current findings. As described above, the ramifications of model simplification in terms of reduced vertical discretization in the uncertainty quantification and data assimilation context more generally is covered by the separate manuscript White et al. (forthcoming). We will nevertheless build on the brief explanation provided in the current manuscript for completeness.

Many thanks again to Mr Turnadge for his helpful comments.

Matt Knowling

References:

1. Gusyev, M., Abrams, D., Toews, M., Morgenstern, U., and Stewart, M.: A comparison of particle-tracking and solute transport methods for simulation of tritium concentrations and groundwater transit times in river water, Hydrology and Earth System Sciences, 18, 3109, 2014

2. Knowling, M. J., White, J. T., and Moore, C. R.: Role of model parameterization in risk-based decision support: An empirical exploration, Advances in Water Resources, 128, 59 – 73, https://doi.org/10.1016/j.advwatres.2019.04.010, 2019

3. Moore, C., and Doherty, J.: Role of the calibration process in reducing model predictive error, Water Resources Research 41(5), 1-14, 2005

4. Morgenstern, U., Begg, J., van der Raaij, R., Moreau, M., Martindale, H., Daughney, C., Franzblau, R., Stewart, M., Knowling, M., Toews, M., Trompetter, V., Kaiser, J., and Gordon, D.: Heretaunga Plains aquifers : groundwater dynamics, source and hydro-chemical processes as inferred from age, chemistry, and stable isotope tracer data, https://doi.org/10.21420/G2Q92G, 2018

5. Schilling, O. S., Cook, P. G., and Brunner, P.: Beyond Classical Observations in Hydrogeology: The Advantages of Including Exchange Flux, Temperature, Tracer Concentration, Residence Time, and Soil Moisture Observations in Groundwater Model Calibration, Reviews of Geophysics, 57, 146–182, https://doi.org/10.1029/2018RG000619, 2019

6. Turnadge, C., and Smerdon, B.D.: A review of methods for modelling environmental tracers in groundwater: Advantages of tracer concentration simulation, Journal of Hydrology, 519, 3674-3689, https://doi.org/10.1016/j.jhydrol.2014.10.056, 2014

7. Watson, T. A., Doherty, J. E. and Christensen, S.: Parameter and predictive outcomes of model simplification, Water Resources Research, http://dx.doi.org/10.1002/wrcr.20145, 2013

8. White, J.T.: A model-independent iterative ensemble smoother for efficient history-matching and uncertainty quantification in very high dimensions, Environmental Modelling and Software, 109, 191-201, https://doi.org/10.1016/j.envsoft.2018.06.009, 2018

9. White, J. T., Knowling, M. J., and Moore, C. R.: Consequences of model simplification in risk-based decision making: An analysis of groundwater-model vertical discretization, Groundwater, accepted subject to minor revisions, forthcoming

---

## Short Comment (SC5) · 19 Oct 2019

We thank Prof. Ty Ferre for his more detailed review and for his positive sentiments. Below we respond to each of his comments.

Prof. Ferre accurately summarizes our intention to challenge by rigorous investigation the widely accepted perspective among hydrologists and modellers that "more data and more diverse data lead to better model forecasts". Our experience extracting information from hydrologic observations through large, complex models—and the challenges associated with doing so—ultimately led us to undertake this study. We hope that our findings spark important conversations, and that these conversations ultimately lead to

[Figure]

improved decision-support modeling practice.

Prof. Ferre echoes our position that increased model complexity may be required to appropriately assimilate both information-rich data, and data of various types, notwithstanding the additional challenges this added model complexity brings. Striking an appropriate balance between complexity and simplicity in the imperfect model-data assimilation context is an area that requires more attention in our opinion.

We are pleased that Prof. Ferre finds the brevity of the case study details to be appropriate. We believe that this comment will be addressed by the availability of additional details regarding the second case study—the Hauraki Plains paired model analyses used for identifying tracer assimilation-induced bias—following publication of White et al. (forthcoming).

We thank Prof. Ferre for his interesting question on the contextualization and measurement of data worth. While there is probably no single "right" measure of data worth from the perspective of water resource managers across the board, we acknowledge that the assessment of data worth based on changes in the second moment (i.e., variance) of a forecast of management interest is unlikely to express the "full picture" from a decision maker's perspective. This is because decision makers need to evaluate forecast PDFs with respect to carefully defined decision thresholds governing management action. For example, if forecast variance is reduced through data acquisition, but this reduction in variance is of no consequence in terms of our ability to test hypotheses (e.g., if the entire PDF lies on one side of the decision threshold), were the collected data *really* worth it, in terms of the specific decision-support context in question? From the manager's perspective, surely not, as the decision was not made any easier following the acquisition of data.

A more decision maker-focused measure of data worth may therefore be to evaluate forecast PDFs (that are conditioned on different observations) with respect to a specific management decision threshold (or multiple decision thresholds as is more likely the

case in practice; e.g., Vilhelmsen and Ferre, 2017). We feel that such a hypothesis testing approach to data worth assessment has significant merit and requires further investigation, notwithstanding that some excellent related works such as Nowak et al. (2012) and Wagner (1999) already exist in the literature. We also feel that identifying optimal monitoring data with respect to "decision difficulty" (e.g., Knowling et al., 2019) has potential.

For the data worth analysis presented in the first case study, it is difficult to anticipate how the specific outcomes of the analysis would be affected through adopting a hypothesis testing approach to data worth. This is due to the fundamental role that a specified decision threshold plays in the assessment of management action success/failure.

We note that robust assessment of data worth in a hypothesis testing framework should ultimately involve non-linear uncertainty quantification such that not only the second moments but also the first moments of forecast PDFs can be assessed with respect to a decision threshold. Such an approach would also allow for the consequences of data assimilation-induced forecast bias in proximity to a decision threshold to be quantified (e.g., where the value of data in terms of variance reduction may be outweighed by its introduction of bias).

More generally, the decision-support context(s) in which the assessment of data worth can be framed is fundamentally important to the current study and more generally. Adopting a decision-support context allows for the prioritization of data collection towards improving the reliability of model forecasts that are used to support management decision-making. Exploring data worth in other contexts can also nevertheless be performed using similar approaches such as FOSM. For example, in the context of aquifer characterization, one can compute the value of, e.g., geophysics data in terms of the reduction in uncertainty associated with key aquifer properties, which may form the basis for improved system process understanding.

We believe that some of the important points above warrant a brief discussion in the

revised manuscript. We will add this to the Discussion and Conclusions section.

We welcome Prof. Ferre to contact us to discuss further some of the ideas above if he is interested.

Matt Knowling

References

1. Knowling, M. J., White, J. T., and Moore, C. R.: Role of model parameterization in risk-based decision support: An empirical exploration, Advances in Water Resources, 128, 59 – 73, doi:10.1016/j.advwatres.2019.04.010, 2019

2. Nowak, W., Rubin, Y., de Barros, F. P. J.: A hypothesis‐driven approach to optimize field campaigns, Water Resources Research, 48, W06509, doi:10.1029/2011WR011016, 2012

3. Vilhelmsen, T. N., and Ferre, T. P. A.: Extending Data Worth Analyses to Select Multiple Observations Targeting Multiple Forecasts, Groundwater, 56, 399 – 412, doi:10.1111/gwat.12595, 2018

4. Wagner, B. J.: Evaluating data worth for ground-water management under uncertainty, Journal of Water Resources Planning and Management, 125(5), doi: 10.1061/(ASCE)0733-9496(1999)125:5(281), 1999

---

## Referee Comment (RC2) · Anonymous Referee #2 · 13 Nov 2019

In the present manuscript, Knowling et al. aim to demonstrate that environmental tracer observations in general are not as informative for groundwater model data assimilation as previously thought because, in their eyes, flow models are typically too wrong for adequate physical representation of tracer behavior. The authors base their conclusions on only two case studies involving groundwater model calibration against only one environmental tracer (i.e. tritium, in one case study using tritium-derived groundwater residence times and in a second case study using tritium concentrations directly). The authors specifically identify errors in groundwater model vertical discretization as a reason for why data assimilation of groundwater model with tritium concentrations is prone to result in biased model predictions.

While a systematic study on this topic is potentially interesting and useful, the present study lacks the necessary rigor in experimental design and standard in scientific reporting to be able to demonstrate what the study aims to demonstrate and to be a valid contribution to HESS.

Shortcomings include: Failure to properly describe (1) the model calibration procedures, (2) the observation data, and (3) the models and assumptions used to derive residence times from tritium concentrations. The authors also fail in properly referencing scientific literature which already demonstrated aspects of the present study. Moreover, misleading statements are made about existing studies, and the general conclusions that were drawn on the value of environmental tracer observations for groundwater model calibration in general are not justified from the results of the simple experimental setup and use of tritium alone. Due to the lack in reporting, it isn't even possible to fully understand, assess or reproduce the findings.

Below I elaborate on some of the shortcomings of the study which I see as reasons for rejecting of this paper.

——— The manuscript lacks key information on model calibration:

The present manuscript doesn't sufficiently explain the observation data, models which were used to derive the different observation types or calibration procedures.

In the first case study, the value of observations of tritium-derived groundwater residence times are compared to the value of groundwater levels and spring discharge observations for the reduction of the predictive uncertainty of spring discharge predictions. However, information about the calibration procedure is not provided, i.e., it isn't clear whether an ensemble-based data assimilation procedure (i.e., the iterative ensemble smoother as mentioned in the abstract), or whether a classic history matching calibration procedure (i.e., based on a weighted, multivariate maximum likelihood estimation procedure as described in a referenced modelling report) is used. Even though in the abstract it is stated that iterative ensemble smoother was used in the present

study, the method isn't explained in the methods section of model study 1.

One can either assume that it was the same as for model study 2, i.e. Iterative Ensemble Smoother. This is suggested by the wording of the abstract and the term 'data assimilation via history matching' (line 117). An Iterative Ensemble Smoother approach, and ensemble-based data assimilation procedures in general, would however make the direct application of linear predictive uncertainty analysis based on FOSM impossible because to the jacobi matrix isn't calculated by these approaches. Or, one could assume that data assimilation was not conducted but instead classic history matching after reading a referenced modelling report (however, Rakowski and Knowling, 2018, is not referenced in the respective model calibration and uncertainty quantification methodology section (2.4)). Using classic history matching would be a contrast to what was stated in the abstract and make the data worth assessment difficult to compare to the findings of modelling case study 2. The authors should also explain in detail what they mean by how the jacobi matrix was populated.

For the second modelling case study, in section 3 after the description of methods and results of model study 1, it is explained that an Iterative Ensemble Smoother with 100 realisations was used. While for model study 1 it was stated that 882 parameters were calibrated, for model study 2 one does not learn how many parameters were calibrated. While for model study 1 there is a referenced modelling report available, the report referenced for model study 2 was not accepted or published at the time of the article submission and therefore not available for checking (on lines 184-185 it is stated: 'The model, and the vertical-discretization simplification analysis, is described in detail in White et al. (forthcoming)' and the said study is listed in the bibliography as 'accepted, subject to minor revisions').

Key information in the calibration procedure is essential when the purpose of the study is to demonstrate the value of different observation types, as the calibration procedure strongly influence the data worth results.

——— The manuscript lacks key information about the used observation data:

Observation data which were used for the modelling study are not provided, even though this is critical information to understand and reproduce the reported findings.

While for model study 1 at least the different observation types which were used are mentioned, for model study 2 it is completely unclear what observations were used alongside tritium. It isn't clear how many observations of tritium, what uncertainty these observations are associated with, and the study which probably contains such information was not accepted at the time of submission and is not available.

Key questions that should be addressed before data worth can be objectively assessed are: What data were used alongside tritium? Is tritium an informative tracer for each of the two given systems, i.e., is the groundwater residence time in both catchments sensitive to tritium? How was tritium analyzed and which equations were used to post-process tritium concentrations into groundwater residence times? How were flux measurements obtained? What is the uncertainty of spring discharge observations? Are the uncertainties comparable to tritium-based residence time uncertainties? What are the weights that were used during calibration and do they reflect the uncertainty of the different observations? None of this is described in the manuscript.

This information is needed for the readers to assess whether the results of the present study are correct and meaningful.

——— The relevance of the authors' findings is over-stated:

It is unclear why it is concluded that tritium is representative of environmental tracers in general. The manuscript lacks an important number of references which have already published similar results on the value of spring discharge or tritium or which have shown, in much more systematic and rigorous experimental approaches, that environmental tracers are highly valuable for groundwater model calibration.

While the title is very broad, i.e.: 'On the assimilation of environmental tracer observations for model-based decision support', the present study does not generally assess the value of environmental tracer data in a data assimilation context. It appears as if only for one of the two modelling studies formal data assimilation has been conducted (however, as outlined in the previous comment, it is not entirely clear what calibration approach was used in the first modelling case study). Furthermore, only one single environmental tracer is used: tritium. Tritium is certainly not reflective of all environmental tracers and for many groundwater systems, tritium is not a useful tracer because groundwater residence times are of an order on which tritium isn't sensitive. The wording of abstract, introduction, discussion and conclusions strongly suggests that the authors believe that their two case studies of tritium are representative of the wider worth of environmental tracer data for groundwater model calibration (e.g., Lines 268-271) : 'We consider this recommendation to be in stark contrast to the common belief that "calibrating to more data improves the model and its predictions". We therefore also consider this recommendation to be of significant implication to decision-support environmental modeling practitioners. It is expected that this finding can be extended to the general approach of assimilating diverse observation types in environmental modeling.'

Tritium is not representative of environmental tracers in general, as it requires more complex mathematical simulation procedures to do its complex decay and production pathways justice. Showing that a simple one-layer model cannot properly represent tritium transport and therefore calibrating it against tritium results in biased predictions is not generating insights representative for environmental tracer value in general. Numerous previous studies have much more systematically analysed and identified the large benefits of environmental tracers for groundwater model calibration in general, but the large majority are not referenced in the present manuscript. Here are a few examples:

Carniato et al. (2015), Highly parameterized inversion of groundwater reactive transport for a complex field site. DOI: 10.1016/j.jconhyd.2014.12.001.

[Figure]

Delsmann et al. (2016), Global sampling to assess the value of diverse observations in conditioning a real-world groundwater flow and transport model. DOI: 10.1002/2014WR016476

Hunt et al. (2006), The importance of diverse data types to calibrate a watershed model of the Trout Lake Basin, Northern Wisconsin, USA. DOI: 10.1016/j.jhydrol.2005.08.005 (cited in the present manuscript)

Rasa et al. (2013), Effect of different transport observations on inverse modeling results: case study of a long-term groundwater tracer test monitored at high resolution. DOI: 10.1007/s10040-013-1026-8 (cited in the present manuscript)

Xu and Gomez-Hernandez (2016): Characterization of non-Gaussian conductivities and porosities with hydraulic heads, solute concentrations, and water temperatures. DOI: 10.1002/2016WR019011

Oehlmann et al. (2015), Reducing the ambiguity of karst aquifer models by pattern matching of flow and transport on catchment scale. DOI: 10.5194/hess-19-893-2015

Masbruch et al. (2014), Hydrology and numerical simulation of groundwater movement and heat transport in Snake Valley and surrounding areas, Juab, Millard, and Beaver Counties, Utah, and White Pine and Lincoln Counties, Nevada. DOI: 10.3133/sir20145103

What is demonstrated in the first modeling case study, i.e., the complicated nature of using residence/travel time observations derived from tritium for groundwater model calibration, is very well known and was already subject of multiple much more systematic and thorough comparisons and reviews, some of which are even referenced in the present manuscript (e.g., Turnadge and Smerdon 2014 (DOI: 10.1016/j.jhydrol.2014.10.056), McCallum et al. 2014 (DOI: 10.1111/gwat.12052) and 2015 (DOI: doi:10.1111/gwat.12237), Schilling et al. 2019 (DOI: 10.1029/2018RG000619), Sanford 2011 (DOI: 10.1007/s10040-010-0637-6)).

All these studies concluded already that it is better to calibrate a flow model against environmental tracer concentrations, or yet even better, direct flux observations, rather than against residence times due to the fact that the simulation of residence times is often faulty due to structural inaccuracies in the numerical groundwater model.

Specifically, the fact that spring discharge observations contain the largest amount of information for spring discharge predictions is neither surprising nor new. Exchange fluxes in general, be it groundwater discharging as spring water or into a surface water body, or surface water infiltrating into the subsurface, have been demonstrated to not only be more valuable data for groundwater model calibration than travel/residence times observations, but also to be much less prone to bias due to straightforward implementation into flow model calibration compared to the more complex physical underpinnings required for groundwater residence times simulations. The authors even reference one study which has demonstrated this systematically in comparison to groundwater residence time observations: Hunt et al. (2006, DOI: 10.1016/j.jhydrol.2005.08.005, already cited in the manuscript) compared the worth of several different flux observations to the worth of hydraulic heads, environmental tracer concentrations and travel time information, and found that groundwater exfiltration onto the surface (providing baseflow of a stream) was the most information rich data type overall, and that many other flux observation types were also more informative than travel time observations.

The authors failed to reference studies which have already demonstrated the high importance of spring discharge more specifically: Masbruch et al. (2014, DOI: 10.3133/sir20145103, not cited in the manuscript) systematically compared the information content of spring discharge to observations of groundwater levels, temperature and environmental tracers, and found that spring discharge observations were the most informative overall data type. A similarly high importance of spring discharge observations was identified by La Vigna et al. (2006, DOI: 10.1007/s10040-016-1393-z, not cited in the manuscript), who systematically elaborated the worth of spring discharge

observations for the calibration of groundwater flow models in comparison to hydraulic head observations. Oehlmann et al. (2015, DOI: 10.5194/hess-19-893-2015, not cited in the manuscript) systematically analysed the calibration of karst groundwater models against observations of spring discharge, groundwater residence times and groundwater levels. They identified that spring discharge observations provide indispensable information for karst groundwater model calibration, but also showed the large information content of residence time observations. The use of all three observation types together was the most beneficial approach for groundwater model parameterisation.

The authors' literature review is unbalanced, misses many key references, and makes incorrect statements about findings of key studies.

---

## Short Comment (SC6) · 18 Nov 2019

We thank the anonymous reviewer for their comments. We believe that their comments can be addressed where appropriate through some minor, yet important additions and clarifications to the manuscript text. The comments also provide us with an opportunity to revisit and expand on some of our findings and recommendations.

We first wish to clarify the following points related to the reviewer's summary of our work:

– Not only do we assess the apparent or "theoretical" information content of environ-

mental tracer observations for decision-support groundwater model data assimilation, as judged by rigorous data worth exploration, we also assess the potential for the assimilation of these data to cause unwanted effects such as forecast bias, by considering model error and using paired complex/simple models. To our knowledge, no other work has examined the assimilation of environmental tracers in the context of the bias-variance trade-off relevant to the use of imperfect models. We feel that this central aspect of our paper has been over-looked by the reviewer. Framing our findings and recommendations in the context of real-world decision-support modeling is fundamentally important to the purpose of our paper. The importance of this aspect was acknowledged by the other reviewers, e.g., "They make a great case that data, even if it inherently carries important information, can be misleading if it is viewed through the lens of an imperfect model."

– We wish to follow-up on the reviewer's comment that "in (our) eyes, flow models are typically too wrong for adequate physical representation of tracer behavior". Recent studies have showed that even seemingly minor model defects can cause significant ill-effects such as bias and uncertainty under-estimation. However, even more importantly, the outcome of these ill-effects depends on the purpose of the modeling analysis. The challenge is therefore to try to avoid these ill-effects in the context of the given modeling analysis and its purpose. We are advocating for careful and forecast-specific model design, to ensure that the rich information contained within tracer data can be properly assimilated. Potential use of more abstract means to assimilate these data into simpler models, is a promising model design option as it alleviates the need for increased model complexity and the costs associated with it. We suggest that this is an area of future work, as discussed in the Discussion and Conclusions section of the manuscript. These recommendations were explicitly valued by the other reviewers, e.g., "In this case, the group makes the point that increased complexity has a place in assimilating more and more varied data while recognizing that this increased complexity has limits for some applications.".

The reviewer's summary suggests that our conclusions are not warranted on the basis of two case studies and the consideration of one environmental tracer. We wish to point out that an exhaustive exploration of how and when environmental tracers can be most usefully assimilated into models represents a research field in itself. The purpose of our paper is to raise and illustrate the following two points: (i) the assimilation of environmental tracer observations may not always be worthwhile, depending on the forecast being made, and (ii) careful model design is central to the ability to assimilate environmental tracer observations into models. We now address each of the reviewer's comments below.

"Lacking information on model calibration"

The reviewer states that "information about the calibration procedure is not provided" for the first case study. This comment reflects a misunderstanding, and therefore a need to improve the presentation of this portion of manuscript. We employ FOSM techniques to quantitatively assess the worth of tritium-derived MRT along with other hydrologic observations by comparing forecast uncertainty changes following the notional data assimilation of different observations (e.g., see Lines 71-72, 75-78 and 128-130). The linearity assumption underpinning FOSM-based analyses means these powerful techniques can be applied to estimate the "theoretical" worth of data (see Lines 71 and 128), as measured by the change in forecast variance that would occur if uncertain parameters were conditioned using different combinations of observations. The efficiency of FOSM is recognized by assuming the action of a model from parameters to outputs can be characterized by a first-order sensitivity matrix. FOSM analyses do not rely on formal history matching or on the pre-existence of a "calibrated model". To be clear, no actual parameter estimation is undertaken as part of the first case study. FOSM techniques have been widely employed for data worth assessment in this notional context in many settings as it enables rapid exploration of the worth of many different combinations of conditional forecast variances in a computationally efficient manner (e.g., Wallis et al., 2014; Zell et al., 2018). We will address this comment by

revising Line 117 (e.g., add "notional", remove "via history matching") and by replacing the "History matching" sub-section heading with "Observations for assimilation".

We feel that these changes will address the reviewer's confusion regarding whether the iterative ensemble smoother was used for the first case study. The ensemble smoother was used only for the second case study, where we explore the potential for model simplification-induced forecast bias and how this may be exacerbated when assimilating tracer concentration observations. To further address this comment, we will make the distinction between the two different data assimilation approaches undertaken for the two case studies more explicit throughout the revised manuscript. We will also add the words "finite differences" after "Jacobian matrix was populated using" on Line 142.

The paired complex/simple model analysis undertaken for the second case study involves formal history matching and non-linear uncertainty quantification (via the iterative ensemble smoother) for various models with varying vertical discretizations. Each of these analyses are performed twice, once with and once without tritium concentration data for assimilation (i.e., Figure 4 is the result of six ensemble smoother experiments). This first-of-its-kind analysis for environmental tracer assimilation allows us to identify biases arising directly from the assimilation of these information-rich observations with imperfect, real-world groundwater models in a decision-support setting. This approach was necessary to explore otherwise invisible biases induced through assimilating tritium data. As discussed above, we feel this critical and novel part of the paper has been over-looked by the reviewer.

While the number of uncertain model parameters used in the second case study (for each of the 7-, 2- and 1-layer models) is provided in the now-published article White et al. (in press), we agree with the reviewer that this constitutes an important detail that, when absent, may obscure some details regarding the second case study. We will therefore add data assimilation details to Section 3.2 and to the Supplementary Information.

"Lacking information about observation data"

The reviewer is correct that the second case study does not contain information regarding observations used for history matching aside from tritium concentration observations. While the other hydrologic observations used for history matching are described in detail in White et al. (in press) (we will also add a summary of these other observations in response to other comments), we note this omission was purposeful in the original manuscript: the second case study does not compare the relative value of different data including tritium observations (as in the first case study). Instead, the second case study investigates an additional and equally-relevant aspect of environmental tracer assimilation concerning differences in the posterior forecast distribution in terms of first-moment (i.e., bias) and second-moment (i.e., variance) characteristics. These differences arise directly from the assimilation of tritium concentration observations using models that are progressively less equipped to assimilate this information. We will nevertheless add these details to the Supplementary Information to address this comment.

While more information regarding the tritium concentration observation data used for history matching in the second case study are presented in White et al. (in press) (and its SI), we agree with the reviewer that these details are warranted here. We will add details regarding the tritium concentration observations to the revised manuscript.

Responses to specific questions regarding data worth interpretations:

– For reasons described above, we will add details to the Supplementary Information regarding the other observations used alongside tritium for assimilation in the second case study.

– Tritium is indeed an "informative tracer" for the hydrologic settings in both case studies. We agree it is important to state this more explicitly. We will add a sentence to the end of the Introduction to address this comment.

– We agree with the reviewer that details regarding how tritium measurements were interpreted are warranted. This was also raised by Mr Chris Turnadge in his review comments—we refer the reviewer to his comments and our responses. We will add details on this to the description of the first case study (i.e., where MRT observations were used).

– Tritium concentrations are not "post-processed" into MRT for either of the case studies. To address this comment, we will add the following text to the revised manuscript for clarity: "The first case study involves the assimilation of tritium-derived MRT observations (following use of lumped-parameter exponential and binary mixing models). The second case study involves the assimilation of tritium concentration measurements through history matching simulated tritium concentration outputs."

– We will indicate in the revised manuscript the field techniques used to measure fluxes.

– For the first case study, the spring discharge measurement uncertainty is assumed to be proportional to the absolute flux magnitude (i.e., a heteroscedastic error model; e.g., Sorooshian, 1980). The assumed measurement uncertainty for MRT observations are also assumed to be proportional to the MRT magnitude, and are generally lower than those of spring discharge observations (primarily reflecting the difference in magnitudes). However, as described on Lines 144-146 and 300-301, in order to approximately account for the role of model error in reducing a model's ability to fit observations (i.e., we should not be fitting observations to a level commensurate with measurement noise given the presence of model error), we use the model-to-measurement residuals as a basis to adjust the uncertainty surrounding observations (e.g., Doherty, 2015).

– Observation weights assigned to different observations used for assimilation indeed reflect uncertainty—specifically epistemic uncertainty (accounting for both measurement and structural sources of uncertainty)—as described above. This is stated explicitly on Line 143.

"Relevance of findings are over-stated"

The purpose of this paper is to show, through two real-world decision-support models, that the assimilation of environmental tracers is not a panacea to all the ills of environmental simulation, and that care in the model design is essential to ensure that the information contained within these tracers is not squandered. While we will endeavour to make this purpose more clear in the text to avoid misinterpretation, we note that the comment that "the relevance of our findings is over-stated" is in direct contrast to the technically detailed comments by Mr Turnadge, who explicitly stated the carefulness with which our conclusions were drawn: "Having re-read the manuscript, I now believe that the authors have been careful to state that their conclusions regarding the applicability of environmental tracer observations are indicative, rather than comprehensive". We have taken care to ensure that all general statements are accompanied by appropriate caveats.

While we agree that a "systematic study on this topic is potentially interesting and useful" (the reviewer's words), a comprehensive analysis into the value or otherwise of assimilating environmental tracers, and the models required to do so, can never be systematic because it will always be context specific. This context reflects a combination of important factors, such as the decision-support quantities of interest, the hydrologic setting, the complexity (or otherwise) of the model, the other available observations, among others.

On the representativeness of tritium.

We do not "conclude" that tritium is representative of tracers in general. Instead, we consider tritium as a representative environmental tracer in the context of the potential outcomes of its assimilation into imperfect models in general. Our consideration of tritium simply reflects that it is one of the most widely used environmental tracers in younger groundwater systems, as were the subject of the two case studies.

The purpose of the paper is not to assess the representativeness of tritium relative to

other tracers. Instead, it is to demonstrate, in general terms, that the usefulness or otherwise of any information-rich data, including environmental tracers, is related to the forecasts being made, and that such data may induce (undetectable) forecast bias, when assimilated into imperfect models. We believe that these issues are relevant and transferrable to the assimilation of environmental tracers in general. This reflects that we consider the primary barrier to appropriate assimilation of tracer data to be the difficulties associated with extracting information from spatially-discrete concentration observations when using upscaled or simplified representations of hydraulic properties within a model that simulates tracer concentrations using the advection-dispersion equation. To the extent that the simulated output corresponding to observed tracer concentration(s) are sensitive to model details or parameters that are "missing" in a simplified model (e.g., White et al., 2014), inappropriate parameter compensation will occur. Then, to the extent that the forecast of management interest is dependent on these biased parameter estimates, the forecast will be biased, potentially leading to resource mismanagement. Therefore, we believe that these factors, which all real-world decision-support model analyses share, will also cause issues for tracer data assimilation and assimilation of diverse data in imperfect models more generally.

To address comments related to how our findings can be applied to other tracers, we will focus on providing more detail in the Discussion and Conclusions section. Despite this, we believe that the larger challenge for the transferability of our work is the specificity of different decision-support contexts, and the infinite spectrum of model design, as we discuss in the Discussion and Conclusions section.

On citing existing literature.

We note that the reviewer states that only "aspects" of our study have been demonstrated in other studies. We agree. However, as discussed in detail above, the aspects that the reviewer is referring to do not encapsulate the thrust of our paper.

We agree with the reviewer that numerous studies have explored the value in environmental tracer data for history matching groundwater models, and we accept that we have not cited every paper on this in our literature review. However, we do not feel that it is necessary to do so, as the literature we cite in the Introduction collectively expresses the state of the science: that studies have "identified the large benefits of environmental tracers for groundwater model calibration in general" (to use the reviewer's words).

However, our study (or at least our first case study) provides an important demonstration of how "variable" the worth of these data may actually be, e.g., in the presence of other data and when making water quantity-related forecasts. We provide a series of detailed explanations for this—we refer the reviewer to Lines 235-254 of the Discussion and Conclusions sections. We feel that such "worked examples" are important for the community to see—this perspective was strongly supported by the other reviewers, e.g., "This is really important work and I feel that HESS is just the right target audience", "this is just the sort of result that could spark important conversations in our field" and "the group takes advantage of its abilities to provide useful guidance for the community" (Prof. Ferre), and "this manuscript provides a valuable and timely contribution towards guidance in the use of subsurface environmental tracers" and "investigations of when benefits may be obtained (or perhaps more importantly, when not) from these additional data types (and therefore specific guidance in their use) has been limited" (Mr Turnadge).

We note that the references suggested by the reviewer do not tackle the entangled issues of model error, data assimilation and predictive reliability—in contrast to our study. For example, following consideration of the references suggested by the reviewer, we consider questions such as: (i) how could these studies explore the potential for forecast bias given that history matching was undertaken using only a single model?; and (ii) how can the observation data responsible for inducing forecast bias through assimilation be identified?. These questions illuminate how our study differs from previous studies. This also provides an insight into the importance of the context is which our

paper is framed—and how this differs to groundwater modeling practice in general terms. We were very careful to make this point clear, e.g., "Modeling for the purpose of decision support is the context in which the remainder of this paper is framed" (Line 29), and "The benefit or otherwise of direct assimilation of tritium concentration data in other decision contexts, or for more general system understanding and conceptual model development, is therefore not the focus of the current study—this study is concerned with a model's ability to "predict" (in two decision-support contexts) rather than "explain" (observed system behavior), as contrasted by Shmueli (2010)." (a revised version of Line 231). Nevertheless, to address this comment, we will add the references provided by the reviewer to the Introduction.

We agree with the reviewer that it has been reported in the literature that it is a preferred approach to simulate tracer concentrations (involving solution of the advective-dispersive equation) and history match to tracer concentrations directly, rather than simulate residence times (involving advective-only particle-tracking schemes) and history match to derived quantities such as MRT. We state this on Line 40-43. However, in real-world modeling, the compounding challenges associated with simulation of tracer concentrations, e.g., computational demands, can complicate the assimilation of tracers. These challenges are why the latter approach is still popular in the industry (e.g., Turnadge and Smerdon, 2014).

It is interesting that the reviewer states that model "structural inaccuracies" generally explain why residence times cannot be reliably simulated. We contend that a significant degree of "structural inaccuracy" will persist, even when dispersive and decay processes are simulated explicitly (i.e., rather than simplified advective-only simulations), as is demonstrated explicitly in the second case study. The difficulty in simulating spatially and temporally distributed tracer concentrations (or more generally, simulating advective-dispersive transport) has been discussed by many (e.g., Zheng and Gorelick, 2003; Riva et al., 2008). Our second case study demonstrates how discretization-related model error combined with assimilation of discrete-point concentration observations can induce considerable biases and ultimately corrupt resource management.

We agree that the specific finding from the first case study that the spring discharge observations are of most "worth" when considering spring discharge forecasts is not surprising. We state this explicitly on Lines 243-245: "The worth of MRT observations relative to various hydraulic potential and discharge observations across the different forecasts are, in general terms, similar to those reported by Zell et al. (2018)". To address this comment, we will add the Hunt et al. (2006) reference to this sentence, and also add a sentence to the Results section (of the first case study) citing the Masbruch et al. (2014), La Vigna et al. (2006) and Oehlmann et al. (2015) references.

References

Doherty, J.E.: PEST and its utility support software: Theory, Watermark Numerical Publishing, 2015

Riva, M., Guadagnini, A., Fernandez-Garcia, A., Sanchez-Vila, X., Ptak, T.: Relative importance of geostatistical and transport models in describing heavily tailed breakthrough curves at the Lauswiesen site. Journal of Contaminant Hydrology 101, 1, 1-13, doi: 10.1016/j.jconhyd.2008.07.004, 2008

Sorooshian, S.: Parameter estimation of rainfall-runoff models with heteroscedastic streamflow errors — The noninformative data case. Journal of Hydrology 52, 1–2, 127-138, doi: 10.1016/0022-1694(81)90099-8, 1981

Turnadge, C. Smerdon, B.D.: A review of methods for modelling environmental tracers in groundwater: advantages of tracer concentration simulation. Journal of Hydrology 519, 3674–3689, doi: 10.1016/j.jhydrol.2014.10.056, 2014

White, J.T., Doherty, J.E., Hughes, J.D.: Quantifying the predictive consequences of model error with linear subspace analysis. Water Resources Research 50, 2, 1152-1173, doi: 10.1002/2013WR014767, 2014

White, J.T., Knowling, M.J., Moore, C.R.: Consequences of model simplification in risk-based decision making: An analysis of groundwater-model vertical discretization, Groundwater, doi: 10.1111/gwat.12957, In Press

Wallis, I., Moore, C., Post, V., Wolf, L., Martens, E., Prommer, H.: Using predictive uncertainty analysis to optimise tracer test design and data acquisition. Journal of Hydrology 515, 191-204, doi: 10.1016/j.jhydrol.2014.04.061, 2014

Zell, W.O., Culver, T.B., Sanford, W.E.: Prediction uncertainty and data worth assessment for groundwater transport times in an agricultural catchment. Journal of Hydrology 561, 1019-1036, doi: 10.1016/j.jhydrol.2018.02.006, 2018

Zheng, C., Gorelick, S.M.: Analysis of solute transport in flow fields influenced by preferential flowpaths at the decimeter scale. Groundwater 41, 2, 142-155, doi: 10.1111/j.1745-6584.2003.tb02578.x, 2003

---

## Author Comment (AC1) · 2 Dec 2019

**Consolidated Replies to Reviewer Comments**

Matthew J. Knowling[1], Jeremy T. White[2], Catherine R. Moore[2], Pawel Rakowski[3], and Kevin Hayley[4]

[1]Corresponding Author: GNS Science, New Zealand; m.knowling@gns.cri.nz
[2]GNS Science, New Zealand
[3]Hawke's Bay Regional Council, New Zealand
[4]Groundwater Solutions Ltd, Australia

December 3, 2019

Here we respond to each comment raised by the reviewers. Comments are shown in *italics* and are followed immediately by our response.

**1  Reply to Short Comment by Ty P. A. Ferre**

*This is another excellent paper from this group. To me, it sits right between academic and applied hydrology. The group has such advanced modeling skills, that they have immediate credibility when they point out limitations based on model-based interpretations. In this paper, they carry their analysis through to the 'value' of data for decision support. They make a great case that data, even if it inherently carries important information, can be misleading if it is viewed through the lens of an imperfect model. Of course, they are all imperfect models. This is really important work and I feel that HESS is just the right target audience. I hope that it inspires continued careful examination of the interaction of data and models for decision support.*

*Well done Knowling et al!*

*Ty Ferre*

We thank Prof. Ty Ferre for his positive and encouraging comments! We acknowledge his significant contribution to the field of decision-support modeling (the field in which this paper is cast). His comment that our study strikes a balance between academic and applied hydrology is pleasing as it reflects our endeavor to tackle problems that are relevant to both researchers and practitioners.

His comments also reiterate that our findings regarding the potential for assimilation of information-rich tracer data to cause ill-effects can be extended beyond just the (imperfect model) tracer data assimilation context (as covered in the Discussion and Conclusions section of the original manuscript).

Thanks again to Prof. Ferre!

**2 Reply to Referee Comment by Ty P. A. Ferre**

*I provided an informal review previously, this serves as a somewhat more detailed formal review:*

*The authors continue to tackle one of the most important areas of applied hydrogeology with novel and insightful tools. Here, they challenge an accepted fact of hydrogeology that more data, and especially more diverse data will lead to better models for decision support. Their counterintuitive finding that isotopic data may have value little of no value for water resources is just the sort of result that could spark important conversations in our field.*

*As with their previous work, this group takes full advantage of their position as leading hydrogeologic modelers to offer constructive criticism for the field. There is no question that this group can build and calibrate large, complex models and that they are as able as anyone to extract information from hydrologic observations. This lends weight to what could otherwise be criticized as a finding based on lack of response. In this case, the group makes the point that increased complexity has a place in assimilating more and more varied data while recognizing that this increased complexity has limits for some applications. Again, the group takes advantage of its abilities to provide useful guidance for the community.*

We thank Prof. Ty Ferre for his more detailed review and for his positive sentiments. Below we respond to each of his comments.

Prof. Ferre accurately summarizes our intention to challenge by rigorous investigation the widely accepted perspective among hydrologists and modellers that "more data and more diverse data lead to better model forecasts". Our experience extracting information from hydrologic observations through large, complex models—and the challenges associated with doing so—ultimately led us to undertake this study. We hope that our findings spark important conversations, and that these conversations ultimately lead to improved decision-support modeling practice.

Prof. Ferre echoes our position that increased model complexity may be required to appropriately assimilate both information-rich data, and data of various types, notwithstanding the additional challenges this added model complexity brings. Striking an appropriate balance between complexity and simplicity in the imperfect model-data assimilation context is an area that requires more attention in our opinion.

*Personally, I appreciate the terse format of presenting the case studies. There may be call for providing more detail as supplemental information. I believe that the accepted White paper supplies these details. But, I will leave that to other reviewers and the editor to comment on whether it is appropriate to provide more detail in this manuscript.*

We are pleased that Prof. Ferre finds the brevity of the case study details to be appropriate. We believe that this comment has been addressed by: (*i*) the availability of additional details regarding the second case study—the Hauraki Plains paired model analyses used for identifying tracer assimilation-induced bias—following publication of White et al. (ress), and (*ii*) the additional case study details added in response to the comments of the other reviewers (see below).

*My only recommendation with regard to the authors approach reflects my own bias. As such, I completely understand if they do not address it in the paper! Regardless, I would like to hear the authors respond to the following question. From the perspective of a water manager or someone else tasked with assessing hydrologic risk, is a statistical reduction in the forecast the right measure of data value? Would the value of tritium, in this case, be viewed differently if decision-making were seen to be based on hypothesis testing of the plausibility of future high loading, for example? More generally, could the authors comment on the importance of considering the decision making context underlying the assessment of data worth?*

*Fantastic work  I look forward to reading more in the series!*

*Ty Ferre*

We thank Prof. Ferre for his interesting question on the contextualization and measurement of data worth (the assessment performed in the first case study). While there is probably no single "right" measure of data worth from the perspective of water resource managers across the board, we acknowledge that the assessment of data worth based on changes in the second moment (i.e., variance) of a forecast of management interest is unlikely to express the "full picture" from a decision maker's perspective. This is because decision makers need to evaluate forecast PDFs with respect to carefully defined decision thresholds governing management action. For example, if forecast variance is reduced through data acquisition, but this reduction in variance is of no consequence in terms of our ability to test hypotheses (e.g., if the entire PDF lies on one side of the decision threshold), were the collected data *really* worth it, in terms of the specific decision-support context in question? From the managers perspective, surely not, as the decision was not made any easier following the acquisition of data.

A more decision maker-focused measure of data worth may therefore be to evaluate forecast PDFs (that are conditioned on different observations) with respect to a specific management decision threshold (or multiple decision thresholds as is more likely the case in practice; e.g., Vilhelmsen and Ferre (2018)). We feel that such a hypothesis testing approach to data worth assessment has significant merit and requires further investigation, notwithstanding that some excellent related works such as Nowak et al. (2012) and Wagner (1999) already exist in the literature. We also feel that identifying optimal monitoring data with respect to "decision difficulty" (e.g., Knowling et al. (2019)) has potential. For the data worth analysis presented in the first case study, it is difficult to anticipate how the specific outcomes of the analysis would be affected through adopting a hypothesis testing approach to data worth. This is due to the fundamental role that a specified decision threshold plays in the assessment of management action success/failure.

We also note that robust assessment of data worth in a hypothesis testing framework should ultimately involve non-linear uncertainty quantification such that not only the second moments but also the first moments of forecast PDFs can be assessed with respect to a decision threshold. Such an approach would also allow for the consequences of data assimilation-induced forecast bias in proximity to a decision threshold to be quantified (e.g., where the value of data in terms of variance reduction may be outweighed by its introduction of bias).

More generally, the decision-support context(s) in which the assessment

of data worth can be framed is fundamentally important to the current study and more generally. Adopting a decision-support context allows for the prioritization of data collection towards improving the reliability of model forecasts that are used to support management decision-making. Exploring data worth in other contexts can also nevertheless be performed using similar approaches such as FOSM. For example, in the context of aquifer characterization, one can compute the value of, e.g., geophysics data in terms of the reduction in uncertainty associated with key aquifer properties, which may form the basis for improved system process understanding.

We welcome Prof. Ferre to contact us to discuss further some of the ideas above if he is interested.

**3 Reply to Short Comment by Chris Turnadge**

*I would like to thank the authors for their invitation to provide comments on this manuscript. I believe this manuscript provides a valuable and timely contribution towards guidance in the use of subsurface environmental tracers to improve the predictive capability of groundwater flow and transport models. Many publications have implored researchers and practitioners to include a range of non-standard observation types such as temporal differences (Peeters et al., 2011), temperatures (Anderson, 2005), isotope concentrations and activities and inferred residence time (or "ages") (Schilling et al., 2019), and/or geophysical data (Hinnell et al., 2010) in model inversion and prediction uncertainty minimisation. However, investigations of when benefits may be obtained (or perhaps more importantly, when not) from these additional data types (and therefore specific guidance in their use) has been limited.*

We thank Mr Chris Turnadge for his detailed and constructive comments. We agree that the guidance provided in the manuscript is timely for the reasons Mr Turnadge describes (and for reasons described in the Introduction and Discussion sections of the manuscript). It was one of our primary motivations to rigorously investigate the calls for increased use of diverse data (e.g., Schilling et al. (2019)) in the context of decision-support modeling. We thank Mr Turnadge for his positive sentiments.

*On first reading of the manuscript I questioned whether the authors were*

*"overreaching" in their conclusions. Having re-read the manuscript, I now believe that the authors have been careful to state that their conclusions regarding the applicability of environmental tracer observations are indicative, rather than comprehensive. More generally though, I do believe the manuscript will benefit from some revisions. Specifically, I would like to provide three major criticisms of the manuscript. These mainly relate to the suitability of the experimental design, in terms of the suitability of testing the hypotheses presented. These are followed by a number of minor criticisms that I believe the authors should also consider addressing. These minor criticisms mostly relate to the interpretation of environmental tracers, or to descriptions provided of the parameterisations of the numerical models used. I hope my comments are helpful to the authors in improving the manuscript and I am more than willing to provide further clarifications off-line, if they are needed.*

We agree that our conclusions are "indicative", and we are pleased Mr Turnadge believes our carefully formulated conclusions are appropriate. We feel that indicative conclusions are really all that can be drawn on the basis of two (or even more) real-world case study example demonstrations. We also consider empirical demonstrations to provide an important adjunct to theoretical demonstrations; we feel empiricism in the presence of inevitable site specifics are important to accompany theoretical investigation.

We believe that Mr Turnadge's comments can be addressed where appropriate through some minor yet important additions and modifications to the manuscript text. His comments also provide us with an opportunity to reiterate and expand on some of our current decision-support modeling perspectives and philosophies. We respond to each of his comments below.

*Major comments:*

*1. Tracer observations are of limited additional value when direct observations of fluxes are already available*

*For the Heretaunga Plains example, I do not believe that the addition of environmental tracer observations (in this case, tritium) when flux observations (in this case, spring discharge) are already available provides an ideal (i.e. fair) test case. Tracer concentrations are proxies for fluxes (either recharge, lateral or discharge) so they are often measured (and subsequently included in model inversion) when direct observations of fluxes are not available. Assessing the value of tracer observations when flux observations are not available would provide a fairer test case and would be of greater interest. For the Heretaunga Plains example, this could be implemented by simply*

*omitting spring discharge observations from all model inversion.*

We consider the first case study to represent a fair and useful test case for the following reasons. First, our results do in fact already show the worth of MRT observations in the absence of spring discharge observations (albeit in the absence of other observations too—therefore representing the case where the maximum worth of MRT observations is apparent with an otherwise "empty" observation dataset). Please see the blue bars in the MRT column of Figure 2.

Second, the Heretaunga Plains case study presented reflects a real-world investigation, i.e., whereby tritium concentrations were measured after and in combination with discharge measurements. It is not the experience of the authors that tracer concentrations are typically only sampled where flux observations are lacking. This would imply that flux and tracer data contain the same information and can therefore be substituted for one another. Contrast this with much literature suggesting the benefits of as many data types and as much data as possible (as acknowledged by Mr Turnadge above).

We also note that in the Discussion and Conclusions section, we provided a detailed description of circumstances by which the apparent worth of MRT observations in the first case study would likely have been higher (see Lines 246-254 of original manuscript).

*2. Tracer observations are of limited additional value when collected upstream of prediction locations*

*For the Hauraki Plains example, subsurface tritium observations were recorded far upstream of nitrate discharge predictions. Since environmental tracers such as tritium integrate information along flow paths, it is intuitive that these tritium observations would contain limited information of value to predictions of nitrate concentration located far downstream. If subsurface tritium concentrations are available from locations in the vicinity of the predictions of interest, then I suggest that it would be more relevant to include these in the case study.*

We regret that this comment appears to reflect a misunderstanding (and therefore a need to improve the communication of the manuscript). While we agree with Mr Turnadges intuition regarding the integration of tracer-derived information along flow paths, we do not show that tritium concentration observations in the second case study are of limited value or "worth" (regardless of where the observations occurred within the basin). In fact, despite that the second case study does not actually explore the worth or information content of tritium observations, the results suggest that the tritium observations

contain "too much" information (or, rather, misinformation when considered through the lens of an imperfect model that lacks parameter receptacles for the information contained in the tritium observations). That is, the sensitivity of the forecast of interest to uncertain parameters that were conditioned by tritium concentration observations led to forecast bias. This occurs due to two factors: (*i*) the "Firth" nitrate-load forecast aggregates flow paths across the entire domain (i.e., this forecast represents the only nitrate flux sink of the system) and in time; and (*ii*) the tritium observations provide insight into spatially and temporally averaged recharge and lateral flux rates in the upgradient portion of the domain, where most of the surfacewater/groundwater exchange occurs. In other words, the bias reflects the information content of the upgradient tritium observations related to averaged upgradient model parameters on which the forecast is sensitive.

To address this comment, we have added the text "This forecast aggregates flow paths across the entire model domain (i.e., it represents the only nitrate-flux sink of the system)." to the Second case study section of the revised manuscript.

We have also added the text "The biases identified reflect the sensitivity of the Firth forecast to uncertain parameters that were conditioned by tritium concentration observation. This occurs due to the spatially integrated nature of the Firth nitrate-load forecast, and because the tritium observations provide insight into spatially and temporally averaged recharge and lateral flux rates in the upgradient portion of the domain, where most of the surface-water/groundwater exchange occurs." to the Results section of the second case study.

*3. Conservativeness and management thresholds when presenting prediction uncertainty*

*For the Hauraki Plains example, I believe that the authors criticise simple modelling approaches without providing any discussion of whether modelled predictions are conservative. Prediction histograms are presented in the absence of a management threshold; in this case, an upper permissible limit for nitrate discharge. I suggest that the authors present a relevant management threshold when presenting prediction uncertainty results. This would allow the authors the additional benefit of exploring whether predictions produced using simple and complex parameterisation approaches were conservative.*

We agree that exploration of the conservativeness or otherwise of simplified model forecast PDFs (the former of which can be viewed as a metric for accepting such a simplified model) is an important undertaking, especially

when performed with respect to a specific management decision threshold. We are pleased to inform Mr Turnadge that we have two papers that explicitly tackle this question—the first in terms of model parameterization (Knowling et al., 2019) and the second in terms of model vertical discretization (White et al., ress).

We therefore address this comment by adding an explicit reference to these manuscripts and how they cover a broader scope than that of the current manuscript in the Discussion and Conclusions section: "We refer the reader to Knowling et al. (2019) and White et al. (ress) for a broader exploration of the consequences of model simplification (in the form of parameterization reduction and vertical-discretization coarsening respectively) in terms of the decision-relevant forecast bias-variance trade-off and its implications for management decision making more generally.".

To further address this comment, we have added the text "(we refer the reader to White et al. (ress) for an exploration of the appropriateness of simpler-discretization models in decision support more generally)" to the Second case study section.

*Minor comments*

*1. Mean residence times*

*Mean residence time (MRTs) values were used to quantify the age of groundwaters. However, MRTs are equivalent to groundwater ages only under very strict assumptions; specifically, requiring highly simplified conceptualisations. The latter are the basis of mixing functions used in lumped parameter models. In most (complex, real-world) cases, MRTs act as a fitting parameter. This is especially true if MRTs are derived from binary mixing models, which arbitrarily blend two mixing models, which generally undermines their physical bases. As a solution, and rather than the use of lumped parameter models to derived mean residence times, where possible I suggest simulating the reactive transport (or at least, non-reactive transport with first order decay) of environmental tracer concentrations (or activities. Admittedly, for some tracers (such as carbon-14) the complexity of reactions may make reactive transport simulation prohibitive. However, the examples presented by the authors already feature combined numerical flow and transport models. Additionally, the authors examples also feature a relatively simple tracer requiring only the simulation of first order decay, so I would assume that reactive transport simulation would be feasible.*

We agree with Mr Turnadge regarding the simplicity and lack of physical basis of LPMs, and the potential benefits of full advective-dispersive (and reactive) transport numerical models for simulating tracer concentrations (as described comprehensively by Turnadge and Smerdon (2014)), notwithstanding their practical limitations, which are amplified in formal decision-support modeling contexts.

Importantly, however, we reiterate that LPM-derived MRT observations are only used in the first case study (in combination with advective-transport simulations); the second study employs full advective-dispersive transport modelling together with a first-order reaction rate to simulate radioactive decay of tritium (see also comment below). Our intention here was to employ both of these "standard practice" tracer modeling techniques as a basis for exploring the ramifications of model tracer-data assimilation, such that the findings can be as useful as possible to industry. The literature reflects the common use of both advective-only particle-tracking simulations (combined with LPM-based "age" observations) and advective-dispersive simulations (combined with tracer concentrations) (e.g., Gusyev et al. (2014)).

We have addressed this comment by more explicitly stating in the Introduction section (where the two case studies are introduced) the different modeling approaches undertaken for the two case studies.

*2. Binary mixing models*

*The authors state that, as part of lumped parameter modelling, binary mixing models (BMMs) were used to derive mean residence times from sub-surface tritium concentrations. BMMs are a linear combination of two other (ideally) physically-based mixing models. I suggest that the authors describe which mixing models were combined using the BMMs, the relative contribution of each, and the physical meaning of the combined result.*

The MRT observations that are considered in the first case study only, were derived using a combination of exponential piston flow models (EPMs) and BMMs (comprising two "parallel" EPMs), as described in detail in Morgenstern et al. (2018).

We have addressed this comment by adding the following text to the First case study section of the manuscript: "Specifically, a combination of exponential piston-flow models (EPMs) and binary-mixing models (BMMs) (that comprise two EPMs) were used. BMMs were employed for wells where long time-series data are available for multiple tracers, and where an adequate fit to different tracer signals could not be obtained on the basis of a single EPM. Relative EPM mixing fractions were specified on the basis of aquifer confinement conditions and well-screen length (mixing fractions of 80 - 95% were applied for wells with a long screen in unconfined conditions, whereas

mixing fractions of 50 - 60% were applied for wells with shorter screens in confined conditions). The reader is referred to Morgenstern et al. (2018) for more details.".

*3. Atmospheric tritium concentrations*

*The authors state that the historic record of atmospheric tritium concentration features a "shape [that allows] for unique interpretation" (lines 69 - 70). It is unclear what the authors mean by the term "unique" in this context. Unlike SF6, it would not be true to state that the historical atmospheric tritium record is consistently monotonic. Tritium values increased initially due to nuclear weapons testing and have declined ever since. In some cases, the historical atmospheric tritium record features more than one peak value after the cessation of nuclear weapons testing. In addition, historic records of atmospheric tritium concentrations at various locations are typically quite noisy, unlike SF6. See Figure 1 in McCallum et al. (2014) for an example of a noisy, two-peak example of a historical atmospheric tritium record. For this example, and following correction for radiometric decay in the subsurface, a tritium observation of 20 TU would correspond to any of four different recharge times (and therefore ages). In comparison, a historical atmospheric SF6 record is also shown by McCallum et al. (2014), which increases monotonically and would therefore permit unique interpretations of ages from measured concentrations. I suggest that the authors could state instead that tritium is a popular tracer for the identification of young age groundwaters (i.e. <70 years old) for the following reasons. Unlike CFCs, it is not affected by microbial degradation or contamination and, unlike SF6, it is not affected by potential subsurface sources. The authors may wish to cite Beyer et al. (2014), who provided a comparison of traditional (e.g. 3H, CFCs, SF6) and emerging (e.g. Halon-1301, SF5CF3) young age tracers.*

We agree with the need to be more specific regarding the reasons why tritium is a often-favoured tracer and indeed why its consideration herein is relevant. We thank Mr Turnadge for his suggested revision to the text. We have revised the manuscript directly following his suggestion.

*4. Non-reactive modelling of contaminant transport*

*The Hauraki Plains model simulated nitrate as a non-reactive constituent. In practice, nitrates in the subsurface are subject to a range of processes: assimilation, nitrification/denitrification, volatilisation, sorption/desorption and retardation (Kendall and Aravena, 2000). For this reason, I suggest that the authors explain why nitrate was not simulated as a reactive constituent in the forward model.*

The second case study does in fact use a first-order decay rate to simulate the process of denitrification reactions. We consider this approach to be "standard" modeling practice.

We have addressed this comment by adding the text "Denitrification and radioactive tritium decay processes are simulated using first-order reaction rates." to the Second case study section of the revised manuscript.

*5. Screen lengths of wells sampled for environmental tracers*

*When interpreting subsurface concentrations of environmental tracers, knowledge of the length of screened sections in sampled groundwater wells is crucial. If lumped parameter models are used, this affects the choice of mixing model. For example, if sampled wells are open holes or fully screened then the exponential mixing, exponential-piston flow, or dispersion models may be appropriate. Alternatively, if sampled wells are partially screened then the partial exponential model may be appropriate. Given the importance of this information to tracer interpretation, I suggest that the authors describe the screen extents of each sampled well. The authors could also state how this information was used to select an appropriate mixing model for lumped parameter modelling, from which mean residence times were calculated.*

We agree with Mr Turnadge regarding the importance of the well screen length for tracer interpretation—especially when using LPMs for interpretation (as is the case for the first case study).

We have addressed this comment by indicating the role that well screen lengths had on the deployment of LPMs to infer MRT in the First case study section, as per our response to minor comment (2). A full description of well details such as depth, screen interval and the corresponding LPM used for MRT interpretation can be found in Morgenstern et al. (2018) (publicly available online, as referenced in the original manuscript). We therefore prefer not to repeat these details in the current manuscript.

*6. Pilot point parameterisation*

*Pilot point parameterisations of the Heretaunga Plains and Hauraki Plains models are not described explicitly in the manuscript. Specifically, it is not clear which parameters were parameterised using this method. I suggest that the authors describe explicitly which model parameters were implemented on a cell-by-cell basis, or using pilot points, zonation or using spatially uniform values, including horizontal and vertical hydraulic conductivity, specific yield/storage, recharge and, for transport models, porosity.*

We have addressed this comment by the following changes.

First, we have added the following text to the First case study section:

"Spatially-distributed parameterization of hydraulic conductivity (horizontal and horizontal/vertical anisotropy ratio), effective porosity, specific storage and specific yield is achieved using pilot points (e.g., Doherty (2003)). Spatially-distributed river-bed and boundary conductance parameters are defined on a reach and zone basis, respectively. We refer the reader to the Supplementary Information for more information. ".

Second, we have added the following text to the Second case study section: "Spatially-distributed parameterization of (horizontal and vertical) hydraulic conductivity, effective porosity, recharge rate, first-order denitrification rate, initial concentration and dispersivity is achieved using a combination of cell-based and zone-based multipliers. Nitrate-loading rate and abstraction well rate is parameterized using cell-by-cell and well-based mulitipliers, respectively. Streamflow-routing (SFR) elements are parameterized on a stream segment basis. We refer the reader to White et al. (ress) for more information.".

*7. Variogram definitions*

*It is not clear whether, for a given model, the same variogram was used to implement pilot point parameterisation for one or many parameter types. For example, I would not expect that the spatial correlation between hydraulic conductivity values to be the same as for recharge rates. I suggest that the authors state explicitly which variogram parameter values (e.g. correlation length, range, sill, nugget) were used to define which model parameter values, and describe the spatial analyses used to quantify spatial correlation between parameter values. Given that the degree of model complexity (particularly in relation to the ability of a model to assimilate observed data) is a key focus of the manuscript, I believe that detailed descriptions of the model parameterisation used are relevant.*

Briefly, for the Heretaunga Plains case study, the same variogram (variogram parameters already defined in manuscript—see Lines 136-139) is used for pilot-point based distributed parameters; no spatial correlation is assumed otherwise. We have addressed this comment by adding "pilot-point based" to Line 137 of the original manuscript. This addition, in combination with the details on parameterization devices added in response to minor comment (6), addresses this comment for the first case study.

For the Hauraki Plains case study, variogram details regarding the various different parameter types have already been described in both White (2018) and White et al. (ress). We therefore address this comment by adding a reference to the additions made in response to minor comment (6): "We

refer the reader to White (2018) and White et al. (ress) for more information on model parameterization and construction of prior parameter covariance matrices".

On a more general note, we agree that variograms could theoretically be defined on a parameter type-specific basis from a physically-based (or perhaps more appropriately a "physically-motivated") parameter standpoint. However, given our recent experience and findings regarding the significant potential for ill-effects (e.g., forecast bias) in real-world decision-support modeling (e.g., Knowling et al. (2019); White et al. (ress)), we tend to consider spatially distributed parameters employed by regional-scale models to be significant abstractions (i.e., from their intended property representation; e.g., Watson et al. (2013)). It follows that questions such as "what is the variogram for spatially distributed recharge bias-correction parameters?" and "how do we represent uncertainty in the variogram model used to describe prior parameter correlation and heterogeneity (i.e., a "hyper-parameter")?" arise when trying to rigorously deal with real-world model error.

*8. Bias and underestimation*

*The authors state that the assimilation of tritium can induce "biased first moments or underestimated second moments" (line 55). I suggest that the authors could unpack this statement by providing simple examples to support this statement, for both bias and underestimation. The authors could also state explicitly the nature of the bias; i.e. whether prediction mean values were under- or overestimated.*

On Line 55 (of the original manuscript), we are making only a general statement that data assimilation through history matching an imperfect model can result in forecast bias (either mean over- or under-estimation) and/or uncertainty underestimation; these ill effects occur as a result of both model simplification and history matching. We cite literature that demonstrate these phenomena. This statement does not relate to tracers or tritium specifically, or any other data in particular. Therefore, no comments can be made at this point as to the nature of bias or variance underestimation. The direction of tritium assimilation-induced bias (i.e., under- or over-estimation) is covered in the Results section of the second case study (although we note that the direction of bias may not be very generalizable between different forecasts and between different sites).

*9. Ensemble size representativeness*

*The authors state that their implementation of the Iterative Ensemble Smoother featured ensemble sizes of 100 (lines 194-195). This value ap-*

*pears to have been selected arbitrarily, likely based on logistical constraints (e.g. forward model and inversion computing times). Was bootstrapping or other representativeness/convergence testing methods used to assess whether an ensemble size of 100 representative, and/or whether ensemble statistics converged as the ensemble size approached 100? I suggest that the authors demonstrate that the ensemble size used was representative.*

The ensemble size was in fact selected on the basis of an approximation of the solution-space dimensionality. This approximation was obtained through a subspace analysis of predictive error variance (Moore and Doherty, 2005). We refer Mr Turnadge to the Supplementary Information of Knowling et al. (2019) for more information on this, including a plot of the singular-value spectrum.

To address this comment, we have added the above-mentioned singular-value spectrum plot to the Supplementary Information, and a supporting sentence to the Second case study section.

*10. Vertical coarsening of model grid*

*The authors provide limited explanation of why vertical coarsening of the model grid led to fewer relatively long flow paths. Since this observation is crucial to the interpretation of the authors results, I suggest that the authors expand their discussion of this key point.*

Fewer relatively long flow paths occur when vertically coarsening the model grid simply due to the aggregation of numerical discretization effects—the flow paths of a coarser-layer model will be a smoother and averaged representation of those derived from a finer-layer model. As described above, the ramifications of model simplification in terms of reduced vertical discretization in the uncertainty quantification and data assimilation context more generally is covered by the separate manuscript White et al. (ress). Nevertheless, we agree that this is an important explanation to support the current findings. We have therefore added the following explanation to the Results section of the second case study in the revised manuscript: "Briefly, this occurs due to the aggregation of numerical discretization effects—the flow paths of a coarser-layer model will be a smoother and averaged representation of those derived from a finer-layer model".

Many thanks again to Mr Turnadge for his helpful comments.

**4 Reply to Anonymous Referee #2**

*In the present manuscript, Knowling et al. aim to demonstrate that environmental tracer observations in general are not as informative for groundwater model data assimilation as previously thought because, in their eyes, flow models are typically too wrong for adequate physical representation of tracer behavior. The authors base their conclusions on only two case studies involving groundwater model calibration against only one environmental tracer (i.e. tritium, in one case study using tritium-derived groundwater residence times and in a second case study using tritium concentrations directly). The authors specifically identify errors in groundwater model vertical discretization as a reason for why data assimilation of groundwater model with tritium concentrations is prone to result in biased model predictions.*

*While a systematic study on this topic is potentially interesting and useful, the present study lacks the necessary rigor in experimental design and standard in scientific reporting to be able to demonstrate what the study aims to demonstrate and to be a valid contribution to HESS. Shortcomings include: Failure to properly describe (1) the model calibration procedures, (2) the observation data, and (3) the models and assumptions used to derive residence times from tritium concentrations. The authors also fail in properly referencing scientific literature which already demonstrated aspects of the present study. Moreover, misleading statements are made about existing studies, and the general conclusions that were drawn on the value of environmental tracer observations for groundwater model calibration in general are not justified from the results of the simple experimental setup and use of tritium alone. Due to the lack in reporting, it isnt even possible to fully understand, assess or reproduce the findings. Below I elaborate on some of the shortcomings of the study which I see as reasons for rejecting of this paper.*

We thank the anonymous reviewer for their comments. We believe that their comments can be addressed where appropriate through some minor, yet important additions and clarifications to the manuscript text. The comments also provide us with an opportunity to revisit and expand on some of our findings and recommendations.

First we wish to clarify the following points related to the reviewers summary of our work:

1. Not only do we assess the apparent or theoretical information content of environmental tracer observations for decision-support groundwater model data assimilation, as judged by rigorous data worth exploration, we

also assess the potential for the assimilation of these data to cause unwanted effects such as forecast bias, by considering model error and using paired complex/simple models. To our knowledge, no other work has examined the assimilation of environmental tracers in the context of the bias-variance trade-off relevant to the use of imperfect models. We feel that this central aspect of our paper has been over-looked by the reviewer. Framing our findings and recommendations in the context of real-world decision-support modeling is fundamentally important to the purpose of our paper. The importance of this aspect was acknowledged by the other reviewers, e.g., "They make a great case that data, even if it inherently carries important information, can be misleading if it is viewed through the lens of an imperfect model" and "This manuscript provides a valuable and timely contribution towards guidance in the use of subsurface environmental tracers to improve the predictive capability of groundwater flow and transport models. Many publications have implored researchers and practitioners to include a range of non-standard observation types ... However, investigations of when benefits may be obtained (or perhaps more importantly, when not) from these additional data types (and therefore specific guidance in their use) has been limited".

2. We wish to follow-up on the reviewers comment that "in (our) eyes, flow models are typically too wrong for adequate physical representation of tracer behavior". Recent studies have showed that even seemingly minor model defects can cause significant ill-effects such as bias and uncertainty under-estimation. However, even more importantly, the outcome of these ill-effects depends on the purpose of the modeling analysis. The challenge is therefore to try to avoid these ill-effects in the context of the given modeling analysis and its purpose. We are advocating for careful and forecast-specific model design, to ensure that the rich information contained within tracer data can be properly assimilated. Potential use of more abstract means to assimilate these data into simpler models, is a promising model design option as it alleviates the need for increased model complexity and the costs associated with it. We suggest that this is an area of future work, as discussed in the Discussion and Conclusions section of the manuscript. These recommendations were explicitly valued by the other reviewers, e.g., "In this case, the group makes the point that increased complexity has a place in assimilating more and more varied data while recognizing that this increased complexity has limits for some applications."

The reviewers summary suggests that our conclusions are not warranted

on the basis of two case studies and the consideration of one environmental tracer. We wish to point out that an exhaustive exploration of how and when environmental tracers can be most usefully assimilated into models represents a research field in itself. The purpose of our paper is to raise and illustrate the following two points: (*i*) the assimilation of environmental tracer observations may not always be worthwhile, depending on the forecast being made, and (*ii*) careful model design is central to the ability to assimilate environmental tracer observations into models.

We now address each of the reviewer's comments in detail below.

- *The manuscript lacks key information on model calibration:*

  *The present manuscript doesnt sufficiently explain the observation data, models which were used to derive the different observation types or calibration procedures. In the first case study, the value of observations of tritium-derived groundwater residence times are compared to the value of groundwater levels and spring discharge observations for the reduction of the predictive uncertainty of spring discharge predictions. However, information about the calibration procedure is not provided, i.e., it isnt clear whether an ensemble-based data assimilation procedure (i.e., the iterative ensemble smoother as mentioned in the abstract), or whether a classic history matching calibration procedure (i.e., based on a weighted, multivariate maximum likelihood estimation procedure as described in a referenced modelling report) is used. Even though in the abstract it is stated that iterative ensemble smoother was used in the present study, the method isnt explained in the methods section of model study 1.*

  *One can either assume that it was the same as for model study 2, i.e. Iterative Ensemble Smoother. This is suggested by the wording of the abstract and the term data assimilation via history matching (line 117). An Iterative Ensemble Smoother approach, and ensemble-based data assimilation procedures in general, would however make the direct application of linear predictive uncertainty analysis based on FOSM impossible because to the jacobi matrix isn't calculated by these approaches. Or, one could assume that data assimilation was not conducted but instead classic history matching after reading a referenced modelling report (however, Rakowski and Knowling, 2018, is not referenced in the respective model calibration and uncertainty quantification methodology section (2.4)). Using classic history matching would be a contrast*

*to what was stated in the abstract and make the data worth assessment difficult to compare to the findings of modelling case study 2. The authors should also explain in detail what they mean by how the jacobi matrix was populated. For the second modelling case study, in section 3 after the description of methods and results of model study 1, it is explained that an Iterative Ensemble Smoother with 100 realisations was used. While for model study 1 it was stated that 882 parameters were calibrated, for model study 2 one does not learn how many parameters were calibrated. While for model study 1 there is a referenced modelling report available, the report referenced for model study 2 was not accepted or published at the time of the article submission and therefore not available for checking (on lines 184-185 it is stated: 'The model, and the vertical-discretization simplification analysis, is described in detail in White et al. (forthcoming)' and the said study is listed in the bibliography as 'accepted, subject to minor revisions'). Key information in the calibration procedure is essential when the purpose of the study is to demonstrate the value of different observation types, as the calibration procedure strongly influence the data worth results.*

The reviewer states that "information about the calibration procedure is not provided" for the first case study. This comment reflects a misunderstanding, and therefore a need to improve the presentation of this portion of manuscript. We employ FOSM techniques to quantitatively assess the worth of tritium-derived MRT along with other hydrologic observations by comparing forecast uncertainty changes following the notional data assimilation of different observations (e.g., see Lines 71-72, 75-78 and 128-130). FOSM analyses do not rely on formal history matching or on the pre-existence of a "calibrated model". To be clear, no actual parameter estimation is undertaken as part of the first case study. FOSM techniques have been widely employed for data worth assessment in this notional context in many settings as it enables rapid exploration of the worth of many different combinations of conditional forecast variances in a computationally efficient manner (e.g., Wallis et al. (2014); Zell et al. (2018)).

We have addressed this comment by revising Line 117 (i.e., add "notionally", remove "via history matching") and by replacing the "History matching" sub-section heading with "Observations for assimilation".

We feel that these changes will address the reviewer's confusion regarding whether the iterative ensemble smoother was used for the first case study. The ensemble smoother was used only for the second case study, where we explore the potential for model simplification-induced forecast bias and how this may be exacerbated when assimilating tracer concentration observations.

To further address this comment, we have made the distinction between the two different data assimilation approaches undertaken for the two case studies more explicit throughout the revised manuscript.

We have also added the words "finite differences" after "Jacobian matrix was populated using" on Line 142 to address this comment.

The paired complex/simple model analysis undertaken for the second case study involves formal history matching and non-linear uncertainty quantification (via the iterative ensemble smoother) for various models with varying vertical discretizations. Each of these analyses are performed twice, once with and once without tritium concentration data for assimilation (i.e., Figure 4 is the result of six ensemble smoother experiments). This first-of-its-kind analysis for environmental tracer assimilation allows us to identify biases arising directly from the assimilation of these information-rich observations with imperfect, real-world groundwater models in a decision-support setting. This approach was necessary to explore otherwise invisible biases induced through assimilating tritium data. As discussed above, we feel this critical and novel part of the paper has been over-looked by the reviewer.

While the number of uncertain model parameters used in the second case study (for each of the 7-, 2- and 1-layer models) is provided in the now-published article White et al. (ress), we agree with the reviewer that this constitutes an important detail that, when absent, may obscure some details regarding the second case study. We have therefore added the following text to Section 3.2 (as well as other data assimilation details; see responses below): "This parameterization approach gives rise to a problem dimensionality (i.e., total number of uncertain parameters) of 141268, 50180 and 29050 for the 7-layer, 2-layer and 1-layer model history-matching experiments, respectively."

- *The manuscript lacks key information about the used observation data:*
  *Observation data which were used for the modelling study are not provided, even though this is critical information to understand and repro-*

*duce the reported findings. While for model study 1 at least the different observation types which were used are mentioned, for model study 2 it is completely unclear what observations were used alongside tritium. It isn't clear how many observations of tritium, what uncertainty these observations are associated with, and the study which probably contains such information was not accepted at the time of submission and is not available.*

The reviewer is correct that the second case study does not contain information regarding observations used for history matching aside from tritium concentration observations. While the other hydrologic observations used for history matching are described in detail in White (2018) and White et al. (ress), we note that this omission was purposeful in the original manuscript: the second case study does not compare the relative value of different data including tritium observations (as is the case in the first case study). Instead, the second case study investigates an additional and equally-relevant aspect of environmental tracer assimilation concerning differences in the posterior forecast distribution in terms of first-moment (i.e., bias) and second-moment (i.e., variance) characteristics. These differences arise directly from the assimilation of tritium concentration observations using models that are progressively less equipped to assimilate this information.

Nevertheless, to address this comment for completeness, we have added details on the observation data used for history matching in the second case study (other than tritium concentration observations), including plots of observation locations in the Supplementary Information: "All history-matching experiments also included observations other than tritium concentrations, such as long-term averaged groundwater levels and surface-water flows and transient surface-water and groundwater nitrate conentrations were also used in the history-matching experiments (see the Supplementary Information for observation locations)".

While more information regarding the tritium concentration observation data used for history matching in the second case study are also presented in White (2018) and White et al. (ress), we agree with the reviewer that these details are warranted here. We have therefore added the following: "Each of these history-matching experiments included 20 tritium concentration observations from the groundwater system (Figure 2) (see also Supplementary Information for observation locations

per model layer)" to the Second case study section.

*Key questions that should be addressed before data worth can be objectively assessed are: What data were used alongside tritium? Is tritium an informative tracer for each of the two given systems, i.e., is the groundwater residence time in both catchments sensitive to tritium? How was tritium analyzed and which equations were used to postprocess tritium concentrations into groundwater residence times? How were flux measurements obtained? What is the uncertainty of spring discharge observations? Are the uncertainties comparable to tritium-based residence time uncertainties? What are the weights that were used during calibration and do they reflect the uncertainty of the different observations? None of this is described in the manuscript. This information is needed for the readers to assess whether the results of the present study are correct and meaningful.*

Our response to each question above are as follows:

- As described above, we have added details to the Supplementary Information regarding the other observations used alongside tritium for assimilation in the second case study.

- Tritium is indeed an "informative tracer" for the hydrologic settings in both case studies. We agree it is important to state this more explicitly. We therefore address this comment by adding ".... in hydrological environments where young groundwater components are decision relevant" to "we focus specifically on the ramifications of assimilating the information contained within tritium concentration observations and tritium-derived mean residence time (MRT) observations for decision support concerning low flow and nutrient transport at the regional scale" in the Introduction section. We have also added the following text to the Introduction (in response to Mr Turnadge's comments): "Tritium is a popular tracer for the identification of relatively young age groundwaters (i.e., <70 years old), for the following reasons: (*i*) unlike CFCs, tritium is not affected by microbial degradation or contamination; and (*ii*) unlike SF6, it is not affected by potential subsurface sources (e.g., Morgenstern and Daughney (2012); Cartwright and Morgenstern (2012); Beyer et al. (2014))".

- We agree with the reviewer that details regarding how tritium measurements were interpreted (for the first case study) are warranted.

This was also raised by Mr Chris Turnadge in his review comments. We have added details on the interpretation of MRT from tritium measurements to the description of the first case study (i.e., where MRT observations were used)—see responses to Mr Turnadge above.

- We have added the following text to the revised manuscript regarding the field techniques used to measure fluxes: "Flux observations are derived using a range of techniques including flow gauging, electrical conductivity and temperature surveys, water isotopic analyses, visual inspections, etc. (Wilding, 2017)".

- The spring discharge measurement uncertainty is assumed to be proportional to the absolute flux magnitude (i.e., a heteroscedastic error model; e.g., Sorooshian (1981)). The assumed measurement uncertainty for MRT observations are also assumed to be proportional to the MRT magnitude, and are generally lower than those of spring discharge observations (primarily reflecting the difference in magnitudes). However, as described on Lines 144-146 and 300-301, in order to approximately account for the role of model error in reducing a models ability to fit observations (i.e., we should not be fitting observations to a level commensurate with measurement noise given the presence of model error), we use the model-to-measurement residuals as a basis to adjust the uncertainty surrounding observations (see, e.g., Doherty (2015)). We address this comment by adding an explicit reference to Appendix A (where this is discussed in more detail) on Line 143: "(see Appendix A)".

- Observation weights assigned to different observations used for assimilation indeed reflect uncertainty—specifically epistemic uncertainty (accounting for both measurement and structural sources of uncertainty)—as described above. This is stated on Line 143.

- *The relevance of the authors findings is over-stated:*

  *It is unclear why it is concluded that tritium is representative of environmental tracers in general. The manuscript lacks an important number of references which have already published similar results on the value of spring discharge or tritium or which have shown, in much more systematic and rigorous experimental approaches, that environmental tracers are highly valuable for groundwater model calibration. While the title*

*is very broad, i.e.: 'On the assimilation of environmental tracer observations for model-based decision support', the present study does not generally assess the value of environmental tracer data in a data assimilation context. It appears as if only for one of the two modelling studies formal data assimilation has been conducted (however, as outlined in the previous comment, it is not entirely clear what calibration approach was used in the first modelling case study). Furthermore, only one single environmental tracer is used: tritium. Tritium is certainly not reflective of all environmental tracers and for many groundwater systems, tritium is not a useful tracer because groundwater residence times are of an order on which tritium isnt sensitive. The wording of abstract, introduction, discussion and conclusions strongly suggests that the authors believe that their two case studies of tritium are representative of the wider worth of environmental tracer data for groundwater model calibration (e.g., Lines 268-271) : "We consider this recommendation to be in stark contrast to the common belief that calibrating to more data improves the model and its predictions. We therefore also consider this recommendation to be of significant implication to decision-support environmental modeling practitioners. It is expected that this finding can be extended to the general approach of assimilating diverse observation types in environmental modeling."*

All comments are addressed in detail in the following responses. First we respond in more general terms.

The purpose of this paper is to show, through two real-world decision-support models, that the assimilation of environmental tracers is not a panacea to all the ills of environmental simulation, and that care in the model design is essential to ensure that the information contained within these tracers is not squandered. While we have endeavored to make this purpose more clear in the text to avoid misinterpretation, we note that the comment that "the relevance of our findings is overstated" is in direct contrast to the technically detailed comments by Mr Turnadge, who explicitly stated the carefulness with which our conclusions were drawn: "Having re-read the manuscript, I now believe that the authors have been careful to state that their conclusions regarding the applicability of environmental tracer observations are indicative, rather than comprehensive" (see our response above). We have taken care to ensure that all general statements are accompanied by appropriate caveats. Prof. Ferre's comments also reflect the more general implications of our findings (i.e., beyond tritium data assimilation): "They make a great case that data, even if it inherently carries important information, can be misleading if it is viewed through the lens of an imperfect model".

While we agree that a "systematic study on this topic is potentially interesting and useful" (the reviewer's words), a comprehensive analysis into the value or otherwise of assimilating environmental tracers, and the models required to do so, can never be systematic because it will always be context specific. This context reflects a combination of important factors, such as the decision-support quantities of interest, the hydrologic setting, the complexity (or otherwise) of the model, the other available observations, among others.

*Tritium is not representative of environmental tracers in general, as it requires more complex mathematical simulation procedures to do its complex decay and production pathways justice. Showing that a simple one-layer model cannot properly represent tritium transport and therefore calibrating it against tritium results in biased predictions is not generating insights representative for environmental tracer value in general. Numerous previous studies have much more systematically analysed and identified the large benefits of environmental tracers for groundwater model calibration in general, but the large majority are not referenced in the present manuscript. Here are a few examples:*

*Carniato et al. (2015), Highly parameterized inversion of groundwater reactive transport for a complex field site. DOI: 10.1016/j.jconhyd.2014.12.001*

*Delsmann et al. (2016), Global sampling to assess the value of diverse observations in conditioning a real-world groundwater flow and transport model. DOI: 10.1002/2014WR016476*

*Hunt et al. (2006), The importance of diverse data types to calibrate a watershed model of the Trout Lake Basin, Northern Wisconsin, USA. DOI: 10.1016/j.jhydrol.2005.08.005 (cited in the present manuscript)*

*Rasa et al. (2013), Effect of different transport observations on inverse modeling results: case study of a long-term groundwater tracer test monitored at high resolution. DOI: 10.1007/s10040-013-1026-8 (cited in the present manuscript)*

*Xu and Gomez-Hernandez (2016): Characterization of non-Gaussian conductivities and porosities with hydraulic heads, solute concentrations, and water temperatures. DOI: 10.1002/2016WR019011*

*Oehlmann et al. (2015), Reducing the ambiguity of karst aquifer models by pattern matching of flow and transport on catchment scale. DOI: 10.5194/hess-19-893-2015*

*Masbruch et al. (2014), Hydrology and numerical simulation of groundwater movement and heat transport in Snake Valley and surrounding areas, Juab, Millard, and Beaver Counties, Utah, and White Pine and Lincoln Counties, Nevada. DOI: 10.3133/sir20145103*

We do not "conclude" that tritium is representative of tracers in general. Instead, we consider tritium as a representative environmental tracer in the context of the potential outcomes of its assimilation into imperfect models in general. Our consideration of tritium simply reflects that it is one of the most widely used environmental tracers in younger groundwater systems (as is the case for our two case studies—see comments above).

The purpose of the paper is not to assess the representativeness of tritium relative to other tracers. Instead, it is to demonstrate that the usefulness or otherwise of any information-rich data, including environmental tracers, is related to the forecasts being made, and that such data may induce (undetectable) forecast bias, when assimilated into imperfect models. We believe that these issues are relevant and transferrable to the assimilation of environmental tracers in general. This reflects that we consider the primary barrier to appropriate assimilation of tracer data to be the difficulties associated with extracting information from spatially-discrete concentration observations when using upscaled or simplified representations of hydraulic properties within a model that simulates tracer concentrations using the advection-dispersion equation. To the extent that the simulated output corresponding to observed tracer concentration(s) are sensitive to model details or parameters that are "missing" in a simplified model (e.g., White et al. (2014)), inappropriate parameter compensation will occur. Then, to the extent that the forecast of management interest is dependent on these biased parameter estimates, the forecast will be biased, potentially leading to resource mismanagement. Therefore, we

believe that these factors, which all real-world decision-support model analyses share, will also cause issues for tracer data assimilation and assimilation of diverse data in imperfect models more generally. Despite this, we believe that the larger challenge for the transferability of our work is the specificity of different decision-support contexts, and the infinite spectrum of model design, as we discuss in the Discussion and Conclusions section.

To address comments related to how our findings can be applied to other tracers, for example, we have added more detail (including some of the text above) to the Discussion and Conclusions section of the revised manuscript.

*What is demonstrated in the first modeling case study, i.e., the complicated nature of using residence/travel time observations derived from tritium for groundwater model calibration, is very well known and was already subject of multiple much more systematic and thorough comparisons and reviews, some of which are even referenced in the present manuscript (e.g., Turnadge and Smerdon 2014 (DOI: 10.1016/j.jhydrol.2014.10.056), McCallum et al. 2014 (DOI: 10.1111/gwat.12052) and 2015 (DOI: doi:10.1111/gwat.12237), Schilling et al. 2019 (DOI: 10.1029/2018RG000619), Sanford 2011 (DOI: 10.1007/s10040-010-0637-6)). All these studies concluded already that it is better to calibrate a flow model against environmental tracer concentrations, or yet even better, direct flux observations, rather than against residence times due to the fact that the simulation of residence times is often faulty due to structural inaccuracies in the numerical groundwater model.*

We agree with the reviewer that it has been reported in the literature that it is a preferred approach to simulate tracer concentrations (involving solution of the advective-dispersive equation) and history match to tracer concentrations directly, rather than simulate residence times (involving advective-only particle-tracking schemes) and history match to derived quantities such as MRT. We state this on Line 40-43 (original manuscript). However, in real-world modeling, the compounding challenges associated with simulation of tracer concentrations, e.g., computational demands, can complicate the assimilation of tracers. These challenges are why the latter approach is still popular in the industry (e.g., Turnadge and Smerdon (2014); Gusyev et al. (2014)).

It is interesting that the reviewer states that model "structural inaccuracies" generally explain why residence times cannot be reliably simulated. We contend that a significant degree of "structural inaccuracy" will persist, even when dispersive and decay processes are simulated explicitly (i.e., rather than simplified advective-only simulations), as is demonstrated explicitly in the second case study. The difficulty in simulating spatially and temporally distributed tracer concentrations (or more generally, simulating advective-dispersive transport) has been discussed by many (e.g., Zheng and Gorelick (2003); Riva et al. (2008))— see also our response to the previous comment. Our second case study demonstrates how discretization-related model error combined with assimilation of discrete-point concentration observations can induce considerable biases and ultimately corrupt resource management.

*Specifically, the fact that spring discharge observations contain the largest amount of information for spring discharge predictions is neither surprising nor new. Exchange fluxes in general, be it groundwater discharging as spring water or into a surface water body, or surface water infiltrating into the subsurface, have been demonstrated to not only be more valuable data for groundwater model calibration than travel/residence times observations, but also to be much less prone to bias due to straightforward implementation into flow model calibration compared to the more complex physical underpinnings required for groundwater residence times simulations. The authors even reference one study which has demonstrated this systematically in comparison to groundwater residence time observations: Hunt et al. (2006, DOI: 10.1016/j.jhydrol.2005.08.005, already cited in the manuscript) compared the worth of several different flux observations to the worth of hydraulic heads, environmental tracer concentrations and travel time information, and found that groundwater exfiltration onto the surface (providing baseflow of a stream) was the most information rich data type overall, and that many other flux observation types were also more informative than travel time observations.*

*The authors failed to reference studies which have already demonstrated the high importance of spring discharge more specifically: Masbruch et al. (2014, DOI: 10.3133/sir20145103, not cited in the manuscript) systematically compared the information content of spring discharge to observations of groundwater levels, temperature and environmental*

*tracers, and found that spring discharge observations were the most informative overall data type. A similarly high importance of spring discharge observations was identified by La Vigna et al. (2006, DOI: 10.1007/s10040-016-1393-z, not cited in the manuscript), who systematically elaborated the worth of spring discharge observations for the calibration of groundwater flow models in comparison to hydraulic head observations. Oehlmann et al. (2015, DOI: 10.5194/hess-19-893-2015, not cited in the manuscript) systematically analysed the calibration of karst groundwater models against observations of spring discharge, groundwater residence times and groundwater levels. They identified that spring discharge observations provide indispensable information for karst groundwater model calibration, but also showed the large information content of residence time observations. The use of all three observation types together was the most beneficial approach for groundwater model parameterisation.*

*The authors' literature review is unbalanced, misses many key references, and makes incorrect statements about findings of key studies.*

We agree that the specific finding from the first case study that the spring discharge observations are of most "worth" when considering spring discharge forecasts is not surprising. We state this explicitly on Lines 243-245: "The worth of MRT observations relative to various hydraulic potential and discharge observations across the different forecasts are, in general terms, similar to those reported by Zell et al. (2018)".

To address this comment, we will add the Hunt et al. (2006) reference to this sentence, and also add a sentence to the Results section of the first case study citing the Masbruch et al. (2014), La Vigna et al. (2016) and Oehlmann et al. (2015) references.

We note that the reviewer states that only "aspects" of our study have been demonstrated in other studies. We agree. However, as discussed in detail above, the aspects that the reviewer is referring to do not encapsulate the thrust of our paper. We agree with the reviewer that numerous studies have explored the value in environmental tracer data for history matching groundwater models, and we accept that we have not cited every paper on this in our literature review. However, we do not feel that it is necessary to do so, as the literature we cite in the

Introduction collectively expresses the state of the science: that studies have "identified the large benefits of environmental tracers for groundwater model calibration in general" (to use the reviewer's words).

In contrast, our study provides an important demonstration of how "variable" the worth of these data may actually be, e.g., in the presence of other data and when making water quantity-related forecasts. We provide a series of detailed explanations for this—we refer the reviewer to Lines 235-254 of the Discussion and Conclusions sections. We feel that such "worked examples" are important for the community to see—this perspective was strongly supported by the other reviewers, e.g., "This is really important work and I feel that HESS is just the right target audience", "this is just the sort of result that could spark important conversations in our field" and "the group takes advantage of its abilities to provide useful guidance for the community" (Prof. Ferre), and "this manuscript provides a valuable and timely contribution towards guidance in the use of subsurface environmental tracers" and "investigations of when benefits may be obtained (or perhaps more importantly, when not) from these additional data types (and therefore specific guidance in their use) has been limited" (Mr Turnadge).

Furthermore, we note that the references suggested by the reviewer do not tackle the entangled issues of model error, data assimilation and predictive reliability—in contrast to our study. For example, following consideration of the references suggested by the reviewer, we consider questions such as: (*i*) how could these studies explore the potential for forecast bias given that history matching was undertaken using only a single model?; and (*ii*) how can the observation data responsible for inducing forecast bias through assimilation be identified? These questions illuminate how our study differs from previous studies. This also provides an insight into the importance of the context is which our paper is framed—and how this differs to groundwater modeling practice in general terms. We were very careful to make this point clear, e.g., "Modeling for the purpose of decision support is the context in which the remainder of this paper is framed" (Line 29), and "The benefit or otherwise of direct assimilation of tritium concentration data in other decision contexts, or for more general system understanding and conceptual model development, is therefore not the focus of the current study—this study is concerned with a models ability to "predict" (in

two decision-support contexts) rather than "explain" (observed system behavior), as contrasted by Shmueli (2010)." (a revised version of Line 231).

**References**

Beyer, M., Morgenstern, U., and Jackson, B. (2014). Review of techniques for dating young groundwater (<100 years) in new zealand. *Journal of Hydrology (New Zealand)*, 53(2):93–111.

Cartwright, I. and Morgenstern, U. (2012). Constraining groundwater recharge and the rate of geochemical processes using tritium and major ion geochemistry: Ovens catchment, southeast australia. *Journal of Hydrology*, 475:137 – 149.

Doherty, J. (2015). *Calibration and uncertainty analysis for complex environmental models - PEST: complete theory and what it means for modelling the real world.* Watermark Numerical Computing.

Doherty, J. E. (2003). Ground water model calibration using pilot points and regularization. *Ground Water*, 41(2):170–177.

Gusyev, M., Abrams, D., Toews, M., Morgenstern, U., and Stewart, M. (2014). A comparison of particle-tracking and solute transport methods for simulation of tritium concentrations and groundwater transit times in river water. *Hydrology and Earth System Sciences*, 18(8):3109.

Hunt, R. J., Feinstein, D. T., Pint, C. D., and Anderson, M. P. (2006). The importance of diverse data types to calibrate a watershed model of the trout lake basin, northern wisconsin, usa. *Journal of Hydrology*, 321(1):286 – 296.

Knowling, M. J., White, J. T., and Moore, C. R. (2019). Role of model parameterization in risk-based decision support: An empirical exploration. *Advances in Water Resources*, 128:59 – 73.

La Vigna, F., Hill, M. C., Rossetto, R., and Mazza, R. (2016). Parameterization, sensitivity analysis, and inversion: an investigation using groundwater modeling of the surface-mined tivoli-guidonia basin (metropolitan city of rome, italy). *Hydrogeology Journal*, 24(6):1423–1441.

Masbruch, M., Gardner, P., and Brooks, L. (2014). Hydrology and numerical simulation of groundwater movement and heat transport in snake valley and surrounding areas, juab, millard, and beaver counties, utah, and white pine and lincoln counties, nevada.

Moore, C. and Doherty, J. E. (2005). Role of the calibration process in reducing model predictive error. *Water Resources Research*, 41(5):1–14.

Morgenstern, U., Begg, J., van der Raaij, R., Moreau, M., Martindale, H., Daughney, C., Franzblau, R., Stewart, M., Knowling, M., Toews, M., Trompetter, V., Kaiser, J., and Gordon, D. (2018). Heretaunga plains aquifers : groundwater dynamics, source and hydrochemical processes as inferred from age, chemistry, and stable isotope tracer data.

Morgenstern, U. and Daughney, C. J. (2012). Groundwater age for identification of baseline groundwater quality and impacts of land-use intensification the national groundwater monitoring programme of new zealand. *Journal of Hydrology*, 456-457:79 – 93.

Nowak, W., Rubin, Y., and de Barros, F. P. J. (2012). A hypothesis-driven approach to optimize field campaigns. *Water Resources Research*, 48(6).

Oehlmann, S., Geyer, T., Licha, T., and Sauter, M. (2015). Reducing the ambiguity of karst aquifer models by pattern matching of flow and transport on catchment scale. *Hydrology and Earth System Sciences*, 19(2):893–912.

Riva, M., Guadagnini, A., Fernandez-Garcia, D., Sanchez-Vila, X., and Ptak, T. (2008). Relative importance of geostatistical and transport models in describing heavily tailed breakthrough curves at the lauswiesen site. *Journal of Contaminant Hydrology*, 101(1):1 – 13.

Schilling, O. S., Cook, P. G., and Brunner, P. (2019). Beyond classical observations in hydrogeology: The advantages of including exchange flux, temperature, tracer concentration, residence time, and soil moisture observations in groundwater model calibration. *Reviews of Geophysics*, 57(1):146–182.

Sorooshian, S. (1981). Parameter estimation of rainfall-runoff models with heteroscedastic streamflow errors the noninformative data case. *Journal of Hydrology*, 52(1):127 – 138.

Turnadge, C. and Smerdon, B. D. (2014). A review of methods for modelling environmental tracers in groundwater: advantages of tracer concentration simulation. *Journal of Hydrology*, 519:3674–3689.

Vilhelmsen, T. N. and Ferre, T. P. (2018). Extending data worth analyses to select multiple observations targeting multiple forecasts. *Groundwater*, 56(3):399–412.

Wagner, B. J. (1999). Evaluating data worth for ground-water management under uncertainty. *Journal of water resources planning and management*, 125(5):281–288.

Wallis, I., Moore, C., Post, V., Wolf, L., Martens, E., and Prommer, H. (2014). Using predictive uncertainty analysis to optimise tracer test design and data acquisition. *Journal of Hydrology*, 515:191–204.

Watson, T. A., Doherty, J. E., and Christensen, S. (2013). Parameter and predictive outcomes of model simplification. *Water Resources Research*.

White, J. T. (2018). A model-independent iterative ensemble smoother for efficient history-matching and uncertainty quantification in very high dimensions. *Environmental Modelling & Software*.

White, J. T., Doherty, J. E., and Hughes, J. D. (2014). Quantifying the predictive consequences of model error with linear subspace analysis. *Water Resources Research*, 50(2):1152–1173.

White, J. T., Knowling, M. J., and Moore, C. R. (in press). Consequences of model simplification in risk-based decision making: An analysis of groundwater-model vertical discretization. *Groundwater, doi: 10.1111/gwat.12957*.

Wilding, T. K. (2017). Heretaunga springs: Gains and losses of stream flow to groundwater on the heretaunga plains.

Zell, W. O., Culver, T. B., and Sanford, W. E. (2018). Prediction uncertainty and data worth assessment for groundwater transport times in an agricultural catchment. *Journal of Hydrology*, 561:1019 – 1036.

Zheng, C. and Gorelick, S. M. (2003). Analysis of solute transport in flow fields influenced by preferential flowpaths at the decimeter scale. *Groundwater*, 41(2):142–155.

---

## Author Response (AR3)

**Consolidated Replies to Second Round of Reviewer Comments**

Matthew J. Knowling[1], Jeremy T. White[2], Catherine R. Moore[2], Pawel Rakowski[3], and Kevin Hayley[4]

[1]Corresponding Author: GNS Science, New Zealand;
m.knowling@gns.cri.nz
[2]GNS Science, New Zealand
[3]Hawke's Bay Regional Council, New Zealand
[4]Groundwater Solutions Ltd, Australia

February 29, 2020

Here we respond to the comments by the Associated Editor and the anonymous reviewer. Comments are shown in *italics* and are followed immediately by our response.

**1    Reply to Associate Editor Dr Fabrizio Fenicia**

*There is one remaining concern by one of the reviewers, which suggests that although the study is valuable, that there are some misleading statements about existing literature, which result into overstating the relevance of the study's findings. I agree with the reviewer that these statements should be removed or rephrased.*

*Please see the detailed reviewer report, and incorporate the relevant references therein.*

As per the responses below, we have addressed the reviewer's comments by revising the manuscript to more explicitly state how our recommendations

relate to those made in the existing literature. We feel these modifications have clarified the novelty of our paper.

**2 Reply to Anonymous Referee #2**

*This is a review of the revised version of "On the assimilation of environmental tracer observations for model-based decision support" by Knowling et al.*

*I have carefully read the revised manuscript and the authors' responses and believe that the revised manuscript will make a valuable contribution to HESS, provided that minor points in the presentation of the results and discussion of existing literature are considered.*

We thank the anonymous reviewer for their careful review of the manuscript and for their encouraging comments.

*My main concerns with the original manuscript were a.) a lack of information and/or unclear descriptions of the modelling/history matching/uncertainty quantification procedures applied in case study 1, b.) a lack of information on observations used for case study 2, and c.) that the relevance of the findings were overstated.*

*The authors have done a good job in addressing concerns a.) and b.) through appropriate revisions, additions and clarifications. With respect to point c.), the manuscript has improved but not sufficiently. The context of the findings is now much clearer than in the original manuscript, owing to the improved description of observation data, history matching procedures, and uncertainty quantification approaches. I find the findings of this study interesting and relevant, i.e., that a.) tritium-based MRT are not adding a tremendous amount of information to an already rich dataset consisting of hydraulic heads and spring discharge observations for a model aimed at predicting spring discharge (case study 1), and that b.) tritium concentration observations can lead to parameter overcompensation and thus forecasting bias when used to calibrate a two-layer model aimed at predicting nitrate load, where in reality the complex hydrogeological situation demands for a much more complex model structure. However, these findings are presented in an overstating manner. On the one hand side, the authors acknowledge in their introduction that it is already widely understood that the value of a specific observation type is highly dependent on the system modeled, the conceptual and numerical model employed, and the forecast being made (among other*

*controlling variables). Multiple studies and review articles which have already made this point are discussed in the introduction. However, on the other hand side, despite acknowledging this previous work, the authors at the same time present their own findings as if these previous studies and reviews do not exist, suggesting that all others unqualifiedly advocate for more use of environmental tracers without knowledge/critical discussion of the potential pitfalls associated with the use of environmental tracers for environmental model calibration. However, this is completely wrong, as the previous studies and reviews all DO discuss various the potential pitfalls associated with many different observation types. The following sentence from the abstract (lines 17-18) is an example of the overstatement of the findings of this study and the obscuring of findings presented in existing literature:*

*'The findings of this study challenge the unqualified advocacy of the increasing use of tracers, and diverse data types more generally, whenever environmental model data assimilation is undertaken with imperfect models'.*

To address this comment, we have revised the above-referenced sentence. It now reads: "The findings of this study challenge the advocacy of the increasing use of tracers, and diverse data types more generally, whenever environmental model data assimilation is undertaken with imperfect models".

While we agree that other studies including those cited in the current manuscript have discussed various pitfalls regarding the assimilation of tracer observation data, one major challenge that has not been confronted in the context of tracer data assimilation prior to our current study (as far as we are aware) is that of the ramifications of model error in terms of forecast bias and variance (which requires paired model analyses). We cover this in the Introduction and Discussion, as well as in the first response document. This is central to the novelty of our paper, as valued by the other reviewers.

*A large number of studies and review articles explored the worth of diverse data types for environmental model calibration in a scientifically sound manner and justifiably demonstrated that diverse data in many contexts contain highly valuable information for the reduction of parameter and forecast uncertainty of environmental models. This sentence, and others alike throughout the manuscript, are inappropriate and oversell the findings of the present study. Another example is the following sentence from the introduction (lines 48-49):*

*'However, the notion that unabated assimilation of diverse data types (including environmental tracers) is always of benefit holds only from a theoretical standpoint'.*

*This sentence is surprising, because it directly follows the discussion of previous studies and reviews that investigate and highlighted pitfalls of using tracer-based observations in environmental model calibration. The apparent 'notion that unabated assimilation of diverse data is always of benefit' does not exist in the referenced literature.*

We reiterate that our study differs in that it is the first (as far as we are aware) to explore the benefit or otherwise of tracer data assimilation into imperfect models in terms of variance *and* bias. Our demonstration of the impact of the "misdirection" of rich tracer-based information due to structural errors using real-world based empirical examples is central to our study's novelty.

We have therefore addressed this comment by revising the above-referenced sentence as follows: "However, the extent to which the assimilation of diverse data types (including environmental tracers) is of benefit has previously been investigated only from a somewhat theoretical standpoint, i.e., neglecting the effects of model error".

To further address this comment, we have added the following sentence to the Introduction: "To the best of the authors' knowledge, this is the first study to explore the benefit or otherwise of the assimilation of tracer data into imperfect models in terms of both forecast bias and variance".

*Another statement of the same kind is found in the discussion and conclusions section:*

*Lines 311-312: 'We consider this recommendation to be in stark contrast to the common belief that calibrating to more data improves the model and its predictions, and therefore of significant implication to decision-support environmental modeling practitioners.'*

We have addressed this comment by revising the above-referenced sentence. It now reads: "We consider this recommendation to be in stark contrast to what we believe is a common view among practitioners that "calibrating to more data improves the model and its predictions"; we therefore consider this recommendation to be of significant implication to decision-support environmental modeling practitioners".

*The many existing studies and reviews on the topic are evidence that there already is a widespread understanding that the worth of a given observation in a given context is observation-, context-, model- and forecast- specific. In the most recent review on the matter, Schilling et al. (2019) highlight specifically with respect to (mean) residence time (RT) and travel time (TT) observations: 'The above findings confirmed that observations of TT and RT*

*can be beneficial for flow model calibration but that their implementation into a flow model calibration routine requires a significant number of assumptions to be made and, therefore, is associated with uncertainties.'*

*The sentence on lines 292-293 is therefore clearly wrong: 'These findings are nevertheless highly relevant in that MRT observations are widely regarded to be of benefit in constraining uncertain model parameters more generally (Schilling et al., 2019)—regardless of the forecast.'*

We have addressed this comment by revising the above-referenced sentence. It now reads: "These findings are nevertheless highly relevant in that MRT observations are widely used and often regarded to be of benefit in constraining uncertain model parameters more generally (Schilling et al., 2019)".

*In contrast to this misleading statement on lines 292-293, Schilling et al. (2019) further noted: 'Also, observations of RTs and TTs contain valuable information about SWGW flow systems, but their implementation into flow model calibration needs to be done with great care, as it is associated with many potential pitfalls. Unlike exchange flux observations, which do not rely on many underlying assumptions and do not require additional processes to be simulated, the successful implementation of observations of RT and TT into flow model calibration is typically based on many underlying assumptions on the conceptual model and on simplified representations in flow models.' Further: 'However, one should exercise due care when choosing to apply a transformation or a simplified process representation in flow models: Depending on the spatial and temporal scale and the nature of the original observation, transformations and simplifications might result in an over-simplification, and calibration might result in a biased model, of which the structural defects cannot be quantified (see Turnadge & Smerdon, 2014). Transformed observations should thus only be considered for cases where direct integration of the untransformed observations is impossible or unfeasible, or where the transformation is not associated with substantial additional uncertainty.'*

*The existing literature on the matter, here represented by the most recent review by Schilling et al. (2019), clearly discuss the pitfalls associated of using diverse and tracer-based data, and particularly of transformed observations such as (mean) residence times. The statements about the current 'notion', 'common belief' and 'unqualified advocacy' made in this manuscript are thus evidently obscuring the findings of existing literature on the matter and over-selling the findings of the present study. This is not only unnecessary, it is*

*inacceptable conduct.*

We have addressed this comment through the revisions detailed above.

However, we reiterate that previous studies—including the review paper of Schilling et al—do not tackle the ramifications of model error in combination with the other controlling factors associated with model-data assimilation. Such treatments of the pitfalls of the tracer and diverse data assimilation are therefore limited in this regard.

*As elaborated above, the findings of the present study are interesting and relevant and will make a valuable contribution to HESS, but the manuscript must not make misleading statements about existing literature in an attempt to overstate the relevance of the study's findings. All the misleading statements highlighted herein must be removed before this article can be considered for final publication. Minor revisions are therefore necessary prior to publication.*

We thank the reviewer for their careful re-review.

[revised manuscript text omitted]